# Mapping the human subcortical auditory system using histology, postmortem MRI and in vivo MRI at 7T

Kevin R Sitek[1,2†*], Omer Faruk Gulban[3†*], Evan Calabrese[4§], G Allan Johnson[4], Agustin Lage-Castellanos[3], Michelle Moerel[3,5], Satrajit S Ghosh[1,2‡], Federico De Martino[3,6‡]

[1]Massachusetts Institute of Technology, Cambridge, United States; [2]Harvard University, Cambridge, United States; [3]Department of Cognitive Neuroscience, Faculty of Psychology and Neuroscience, Maastricht University, Maastricht, Netherlands; [4]Duke University, Durham, United States; [5]Maastricht Centre for Systems Biology, Faculty of Science and Engineering, Maastricht University, Maastricht, Netherlands; [6]Center for Magnetic Resonance Research, University of Minnesota, Minneapolis, United States

**\*For correspondence:**
ksitek@mit.edu (KRS);
faruk.gulban@
maastrichtuniversity.nl (OFG)

[†]These authors contributed equally to this work
[‡]These authors also contributed equally to this work

**Present address:** [§]Department of Radiology, University of California, San Francisco, United States

**Competing interests:** The authors declare that no competing interests exist.

**Abstract** Studying the human subcortical auditory system non-invasively is challenging due to its small, densely packed structures deep within the brain. Additionally, the elaborate three-dimensional (3-D) structure of the system can be difficult to understand based on currently available 2-D schematics and animal models. Wfe addressed these issues using a combination of histological data, post mortem magnetic resonance imaging (MRI), and in vivo MRI at 7 Tesla. We created anatomical atlases based on state-of-the-art human histology (BigBrain) and postmortem MRI (50 μm). We measured functional MRI (fMRI) responses to natural sounds and demonstrate that the functional localization of subcortical structures is reliable within individual participants who were scanned in two different experiments. Further, a group functional atlas derived from the functional data locates these structures with a median distance below 2 mm. Using diffusion MRI tractography, we revealed structural connectivity maps of the human subcortical auditory pathway both in vivo (1050 μm isotropic resolution) and post mortem (200 μm isotropic resolution). This work captures current MRI capabilities for investigating the human subcortical auditory system, describes challenges that remain, and contributes novel, openly available data, atlases, and tools for researching the human auditory system.
DOI: https://doi.org/10.7554/eLife.48932.001

## Introduction

Understanding the structure of the human subcortical auditory pathway is a necessary step to research its role in hearing, speech communication, and music. However, due to methodological issues in human research, most of our understanding of the subcortical (thalamic, midbrain, and brainstem) auditory pathway arises from research conducted in animal models. This might be problematic because, while the organization of the auditory pathway is largely conserved across mammalian species (*Malmierca and Hackett, 2010*; *Schofield, 2010*), the form and function of each structure may not be analogous (*Moore, 1987*). In this paper, we show that three human imaging modalities – histology, postmortem magnetic resonance imaging (MRI), and in vivo MRI at ultra high-field (7 Tesla) – can identify the structures of the subcortical auditory pathway at high spatial resolution (between 50 and 1100 μm).

Although MRI has become increasingly powerful at imaging deep brain structures, anatomical investigation of the human subcortical auditory pathway has been primarily conducted in postmortem tissue dissection and staining. *Moore (1987)* stained both myelin and the cell bodies of subcortical auditory structures in four postmortem human brainstem samples and compared them to the analogous structures in cats (a common model for auditory investigations at the time). Later investigations from the same group *Moore et al. (1995)* used myelin and Nissl cell body staining to investigate the timeline of myelination in human auditory brainstem development. More recently, *Kulesza (2007)* stained six human brainstems for Nissl substance, focusing on the superior olivary complex, finding evidence of a substructure (the medial nucleus of the trapezoid body) whose existence in the human auditory system has been debated for decades.

Advances in post-mortem human MRI allow for investigating three-dimensional (3-D) brain anatomy with increasingly high resolution (100 μm and below). This points to 'magnetic resonance histology' (*Johnson et al., 1993*) as a promising avenue for identifying the small, deep subcortical auditory structures. However, to the best of our knowledge, postmortem MRI has not been utilized within the subcortical auditory system, although it has provided useful information about laminar structure in the auditory cortex (*Wallace et al., 2016*).

To study the subcortical auditory system in living humans, MRI is the best available tool due to its high spatial resolution. Anatomical in vivo MRI investigations of the human subcortical auditory pathway so far have focused on thalamic nuclei (*Devlin et al., 2006*; *Moerel et al., 2015*), and the identification of the acoustic radiations between the auditory cortex and medial geniculate nucleus of the thalamus with diffusion-weighted MRI tractography (*Devlin et al., 2006*; *Behrens et al., 2007*; *Javad et al., 2014*; *Maffei et al., 2018*). The inferior colliculus of the midbrain can also be identified using anatomical MRI—for instance, *Tourdias et al. (2014)* and *Moerel et al. (2015)* show the inferior colliculus using short inversion time T1-weighted anatomical MRI at 7 Tesla, although neither investigation focused on anatomical segmentation of the inferior colliculus. Due to their small size and deep locations, identification of more caudal subcortical structures-the superior olivary complex and cochlear nucleus-remain challenging with in vivo anatomical MRI.

Although lower spatial resolution than anatomical MRI, functional MRI (fMRI) has been used to investigate the relevance of subcortical processing of auditory information in humans, but it has been limited by the small size of the structures involved and the relatively low resolution attainable at conventional field strengths (3 Tesla and below) (*Guimaraes et al., 1998*; *Harms and Melcher, 2002*; *Griffiths et al., 2001*; *Hawley et al., 2005*). These acquisitions required trade-offs, such as low through-plane resolution (7 mm) in exchange for moderate in-plane resolution (1.6 mm), and in some cases researchers synchronized image collection to the cardiac cycle in order to overcame the physiological noise associated with blood pulsation in the brainstem (*Guimaraes et al., 1998*; *Sigalovsky and Melcher, 2006*).

More recent advances in MRI, especially the increased signal-to-noise ratio (SNR) available at ultra-high magnetic fields (7 Tesla and above), have enabled higher resolution functional imaging of subcortical structures and more advanced localization of human auditory subcortical structures as well as their functional characterization. Using MRI at 7 Tesla (7T), *De Martino et al. (2013)* and *Moerel et al. (2015)* collected relatively high-resolution (1.1–1.5 mm isotropic) fMRI with an auditory paradigm to identify tonotopic gradients in the inferior colliculus and medial geniculate nucleus. In these studies, high isotropic resolution and SNR provided an opportunity to investigate auditory responses throughout the subcortical auditory system.

Despite the methodological advances in investigating the human brain, a systematic comparison of their capabilities for imaging the subcortical auditory system has not yet been undertaken. Here, we use publicly available histological data (*Amunts et al., 2013*) to segment the main nuclei along the subcortical auditory pathway. Using state-of-the-art acquisition and analysis techniques, we evaluate the ability to identify the same structures through postmortem anatomical MRI, through functional MRI using natural sounds, and through estimating the connectivity between subcortical auditory structures with postmortem and in vivo diffusion MRI tractography. To compare the histological, postmortem, and in vivo data, we project all images to MNI common reference space (*Fonov et al., 2009*; *Fonov et al., 2011*). Finally, to facilitate dissemination of our results, we have made the postmortem anatomical data, in vivo functional and diffusion data, and the resulting atlases publicly available.

Where histology provides ground truth information about neural anatomy, we show that post mortem MRI can provide similarly useful 3-D anatomical information with less risk of tissue damage and warping. We also show that in vivo functional MRI can reliably identify the subcortical auditory structures within individuals, even across experiments. Overall, we found that each methodology successfully localized each of the small structures of the subcortical auditory system, and while known issues in image registration hindered direct comparisons between methodologies, each method provides complementary information about the human auditory pathway.

## Results

### Definition of a subcortical auditory atlas from histology

To obtain a spatially accurate reference for all the subcortical auditory structures, we manually segmented publicly available histological data (100 μm version of the BigBrain 3-D Volume Data Release 2015 in MNI space from https://bigbrain.loris.ca; *Amunts et al., 2013*).

Upon inspecting this dataset, we noticed that the area around the inferior colliculus was incorrectly transformed into MNI space. This was causing the colliculi to be larger and more caudal than in the MNI reference brain (Figure 7, second and third panels). Thus, our first step was to correctly register the area around the colliculi (Figure 7, fourth panel; see Materials and methods for details on the correction procedure).

The results of our BigBrain subcortical auditory segmentation in corrected MNI space are reported in *Figure 1* together with schematics redrawn from *Moore (1987)* (for the cochlear nucleus, superior olivary complex, and inferior colliculus) and the Allen Human Brain Atlas (*Hawrylycz et al., 2012*; *Ding et al., 2016*) (for the medial geniculate body). These schematics were used as reference during the segmentation. The 3-D rendering of the segmented structures highlighting the complex shape of the cochlear nucleus and superior olivary complex is also presented in *Figure 1*. The rendering is presented from a posterior lateral view in order to compare it with the Gray's Anatomy, Plate 719 (*Gray and Lewis, 1918*).

### Postmortem MRI

#### Postmortem MRI atlas of the human subcortical auditory system

Magnetic resonance histology—that is the study of tissue at microscopic resolution using MRI—provides several unique advantages over conventional histology: (1) it is non-destructive; (2) it suffers minimal distortion from physical sectioning and dehydration; (3) it yields unique contrast based on water in the tissue and how it is bound (e.g. diffusion); and (4) it produces 3-D data. These advantages make it an ideal medium for visualizing the 3-D organization of the deep brain structures (*Johnson et al., 1993*). To delineate the subcortical auditory structures with MR histology, we acquired 50 μm isotropic voxel size 3-D gradient echo (GRE) MRI on a human postmortem brainstem and thalamus (described previously in *Calabrese et al., 2015*; see Materials and methods for additional details). These data are presented in *Figure 2* (second column) after transformation to MNI space and resampling to 100 μm isotropic resolution (see Materials and methods section for details). The postmortem MRI data are presented together with the histological data for comparison (first column).

Based on our segmentations of the subcortical auditory structures in the postmortem MRI data, the resulting 3-D model is presented in *Figure 2*. A volumetric quantification of the identified structures (in the BigBrain and postmortem MRI) is reported in *Table 1* and the overlap between the segmentations computed after projection in MNI space are reported in Table 2 (as inset in *Figure 2*).

### 3-D connectivity map of the human subcortical auditory system from postmortem diffusion MRI

Identifying the connectivity between subcortical auditory nuclei is crucial for understanding the structure of the pathway. However, methods for tracing neuronal pathways that are available in other animal models are generally not available in human studies, even post mortem. Diffusion-weighted MRI (dMRI) can be used to measure the orientation and magnitude of molecular motion and infer patterns of white matter in brain tissue (both post mortem and in vivo). Using 200 μm diffusion-weighted MRI data acquired on the same post mortem sample (see Materials and methods for

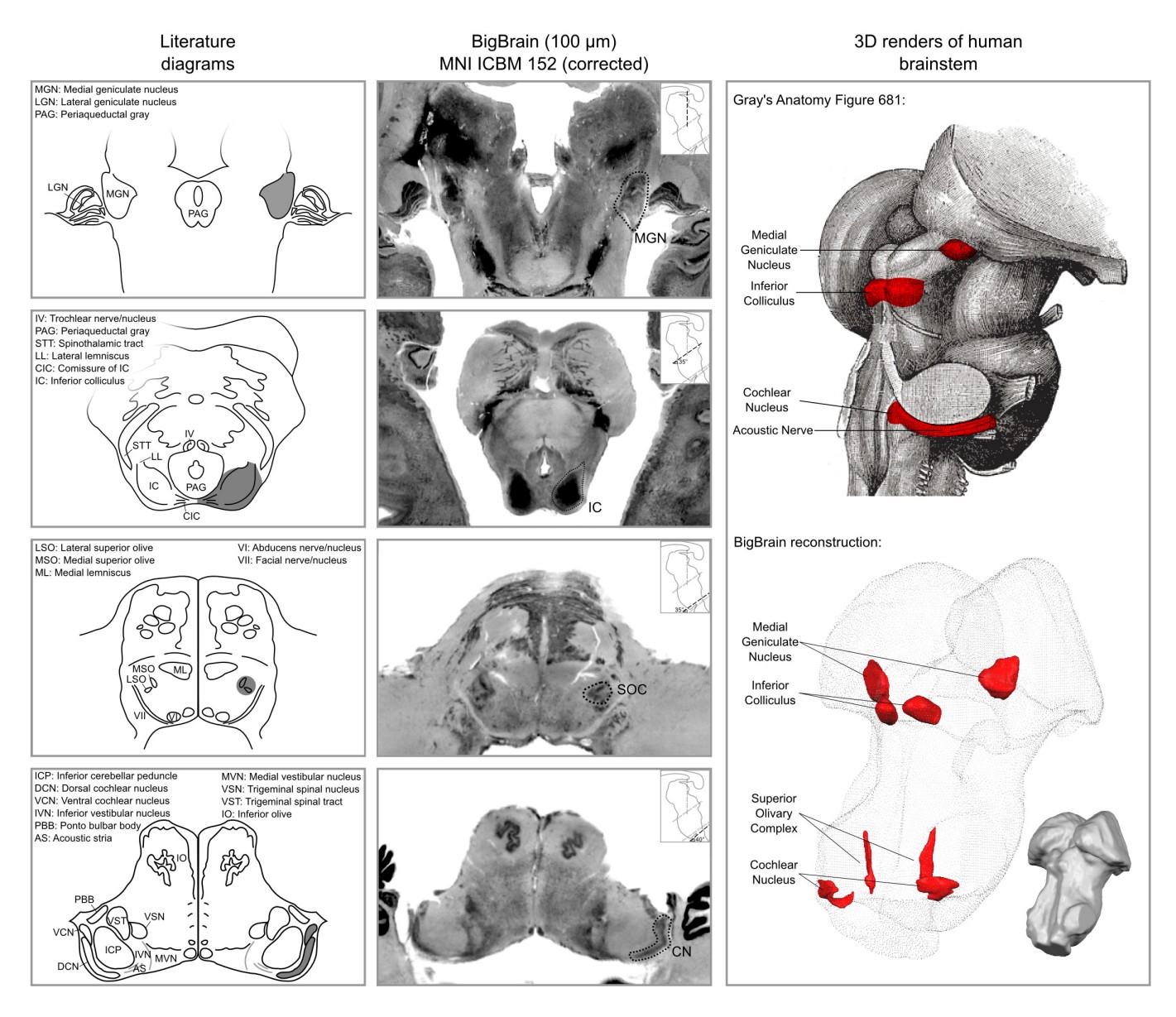

**Figure 1.** Literature diagrams (left columns) redrawn from *Moore (1987)* for the cochlear nucleus (CN), superior olivary complex (SOC), inferior colliculus (IC) and from the Allen Human Brain Atlas (*Hawrylycz et al., 2012*) for the medial geniculate body (MGB) compared to similar cuts from histology (BigBrain) in MNI (central column) and 3-D reconstructions of the segmented structures from the histology (bottom right column). The auditory structures are highlighted in gray in the left column, by a dotted line in the central column and in red on the modified Gray's anatomy Plate 719 (*Gray and Lewis, 1918*) and rendered as solid red surface meshes within the surface point cloud render of BigBrain MNI brainstem (right column). See Figure 9 for 3-D animated videos of these auditory structures.

DOI: https://doi.org/10.7554/eLife.48932.002

details), we modeled diffusion orientations and estimated likely connectivity pathways (or stream-lines) using tractography. Constraining the streamlines to only those that pass through auditory structures (as identified from the anatomical MRI data and dilated 500 µm to include adjacent white matter), we visualized the connectivity map of the subcortical auditory pathway in *Figure 3*, left panel.

Connectivity closely resembles the expected pattern of the human subcortical auditory wiring. In particular, streamlines predominantly pass through the lateral lemniscus, the primary subcortical

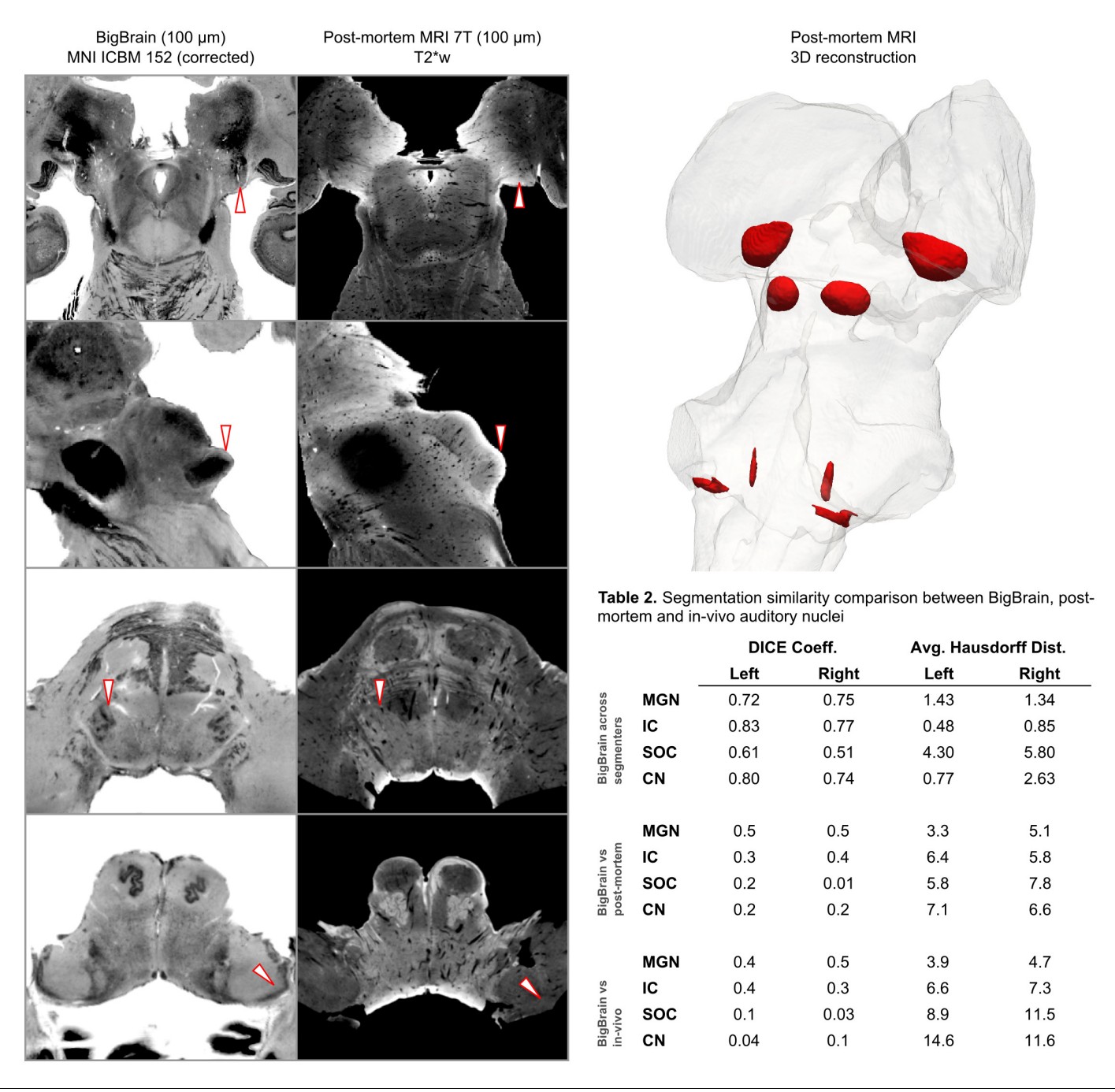

**Table 2.** Segmentation similarity comparison between BigBrain, post-mortem and in-vivo auditory nuclei

| | | DICE Coeff. | | Avg. Hausdorff Dist. | |
|---|---|---|---|---|---|
| | | Left | Right | Left | Right |
| BigBrain across segmenters | MGN | 0.72 | 0.75 | 1.43 | 1.34 |
| | IC | 0.83 | 0.77 | 0.48 | 0.85 |
| | SOC | 0.61 | 0.51 | 4.30 | 5.80 |
| | CN | 0.80 | 0.74 | 0.77 | 2.63 |
| BigBrain vs post-mortem | MGN | 0.5 | 0.5 | 3.3 | 5.1 |
| | IC | 0.3 | 0.4 | 6.4 | 5.8 |
| | SOC | 0.2 | 0.01 | 5.8 | 7.8 |
| | CN | 0.2 | 0.2 | 7.1 | 6.6 |
| BigBrain vs in-vivo | MGN | 0.4 | 0.5 | 3.9 | 4.7 |
| | IC | 0.4 | 0.3 | 6.6 | 7.3 |
| | SOC | 0.1 | 0.03 | 8.9 | 11.5 |
| | CN | 0.04 | 0.1 | 14.6 | 11.6 |

**Figure 2.** BigBrain—7T postmortem MRI image comparisons. Histological data (BigBrain) (left column) and T2*-weighted postmortem MRI data (100 μm - central column) in MNI space. Panels from bottom to top are chosen to highlight subcortical auditory structures (CN [bottom] to MGB [top]). Arrows (white with red outline) indicate the location of the subcortical auditory nuclei. The 3-D structures resulting from the segmentation of the postmortem data is presented on the top right panel. Table 2 quantifies (using DICE coefficient and average Hausdorff distance) the agreement (in MNI space) for all subcortical structures between: (1) segmentations performed on the BigBrain dataset by the two raters (KRS and OFG) [top]; (2) segmentations obtained from the BigBrain dataset and from the post mortem MRI data [middle]; (3) segmentations obtained from the BigBrain dataset and from in vivo functional MRI data [bottom]). See Figure 9 for 3-D animated videos of these auditory structures.
DOI: https://doi.org/10.7554/eLife.48932.003

**Table 1.** Comparisons between the volume (mm³) of auditory subcortical structures reported in the literature (***Glendenning and Masterton, 1998***) and the volume obtained in our BigBrain segmentation (in MNI space), post mortem MRI data segmentation and in vivo functional clusters (defined based on voxels that are significant in at least three, four, or five participants out of the 10 included in Experiment 1).

|  | Literature | BigBrain | Post mortem | In vivo (thr=3) | In vivo (thr=4) | In vivo (thr=5) |
|---|---|---|---|---|---|---|
| **CN** | 46 | 32 | 11 | 54 | 24 | 11 |
| **SOC** | 7 | 6 | 4 | 124 | 63 | 29 |
| **IC** | 65 | 63 | 73 | 263 | 189 | 146 |
| **MGN** | 58 | 75 | 134 | 304 | 207 | 152 |

DOI: https://doi.org/10.7554/eLife.48932.004

auditory tract. Additional streamlines run through the brachium of the inferior colliculus, connecting the inferior colliculus with the medial geniculate of the thalamus. Many streamlines then course rostrally toward the auditory cortex (not present in this specimen).

At the caudal extent of the lateral lemniscus, streamlines pass through the superior olivary complex. Streamlines also run through the root of CNVIII. In total, each expected step along the subcortical auditory pathway is represented in this connectivity map.

***Figure 3*** (top right panel) shows the percentage of total streamlines connecting each of the subcortical auditory structures as estimated from this postmortem diffusion MRI sample. Overall, connections tend to be between ipsilateral structures, with weak connectivity to contralateral structures other than commissural connections to the contralateral homolog (except for between the cochlear nuclei). Still, the majority of streamlines pass through just one region (shown along the diagonal).

To investigate the relationship between streamline connectivity and ROI definition strictness, we conducted two additional analyses. In ***Figure 3***, we dilated the anatomical ROIs by 500 µm (2.5 voxels at 200 µm resolution), thereby including nearby white matter tracts (as well as adjacent subcortical structures). In contrast, ***Figure 3—figure supplement 1*** shows streamlines based on the anatomical ROIs without dilation to account for white matter. As regions were defined as the core nuclei in the anatomical MRI, they largely exclude white matter tracts (such as the lateral lemniscus and brachium of the inferior colliculus), leading to much sparser connectivity between subcortical auditory nuclei.

Next, we resampled the diffusion MRI images to an in vivo-like resolution (1.05 mm isotropic). We again estimated fiber ODFs using CSD and estimated white matter connections with deterministic tractography. Using the (undilated but downsampled) anatomically defined ROIs as tractography waypoints, we can visualize streamline estimates connecting subcortical auditory structures (***Figure 3—figure supplement 2***). Similar to the dilated ROI connectivity estimates, we see greater ipsilateral connectivity estimates between structures, particularly between left structures.

## Vasculature representations from postmortem MRI

Because T2*-weighted GRE imaging is sensitive to blood vessels, we processed our anatomical MR image to highlight brainstem vasculature (Figure 5, right column, base image). These 3-D vasculature images bear striking resemblance to post mortem data acquired with a stereoscopic microscope after full clearing method (see ***Duvernoy, 2013*** for detailed diagrams of human brainstem vasculature). These vasculature images in the MNI space can be helpful to understand the nature of the in vivo functional signals (see next section).

## In vivo MRI

We next sought to identify the structures and connections of the human subcortical auditory system in living participants. By leveraging the increase signal and contrast to noise available at ultra-high magnetic fields (7 Tesla) (***Vaughan et al., 2001***; ***Uğurbil et al., 2003***; ***Ugurbil, 2016***), we collected high-resolution anatomical (0.7 mm isotropic), diffusion-weighted (1.05 mm isotropic; 198 diffusion gradient directions across three gradient strengths) and functional (1.1 mm isotropic) MRI in ten participants (see Materials and methods for details). Leveraging the increased SNR available at high fields, we aimed to collect data that would allow a functional definition of the auditory pathway in

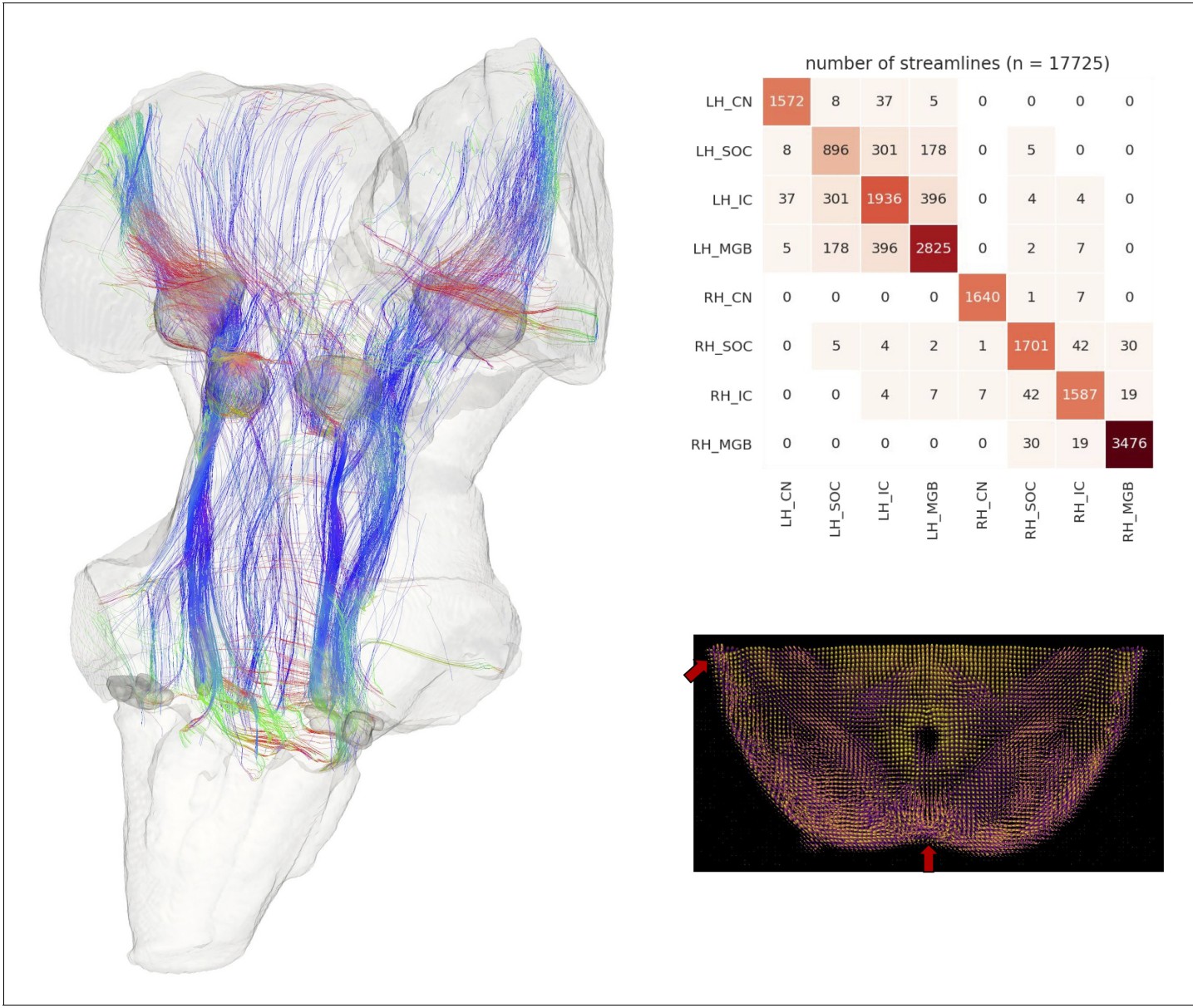

**Figure 3.** Postmortem diffusion MRI tractography. Left: streamlines passing through subcortical auditory structures, defined from 50 μm post mortem MRI in the same specimen, warped to 200μm isotropic diffusion image space and dilated 2.5 voxels (500 μm) to include neighboring white matter. Colors represent the local orientation at each specific point along the streamline: blue is inferior-superior, red green is anterior-posterior, and red is left-right. Ten percent of streamlines are represented in this image. A rotating animation is available in the online resources. Top right: Connectivity heatmap of subcortical auditory structures. Bottom right: Diffusion orientation distribution functions (ODFs) for each voxel; axial slice at the level of the rostral inferior colliculus (IC), including the commissure of the IC (bottom center arrow) and brachium of the IC (top left arrow). A video of the streamlines is available online: https://osf.io/kmbp8/.
DOI: https://doi.org/10.7554/eLife.48932.005

The following video and figure supplements are available for figure 3:

**Figure supplement 1.** Postmortem tractography with undilated ROIs.
DOI: https://doi.org/10.7554/eLife.48932.006
**Figure supplement 2.** Postmortem tractography using data downsampled to in vivo resolution (1.05 mm).
DOI: https://doi.org/10.7554/eLife.48932.007
**Figure 3— video 1.** 360˚ rotation video of postmortem streamlines.
DOI: https://doi.org/10.7554/eLife.48932.008

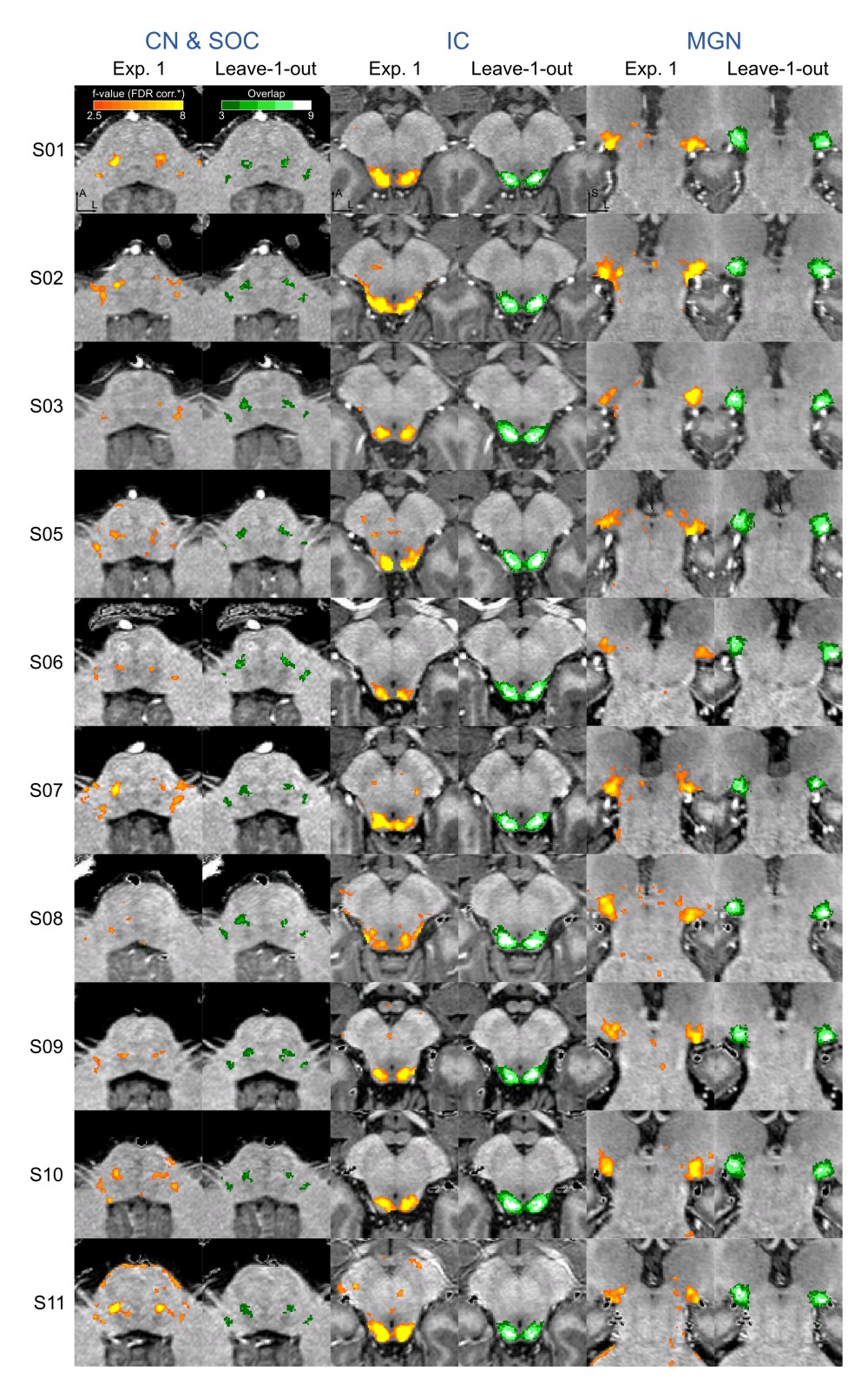

**Figure 4.** Single subject functional activation maps obtained from Experiment one thresholded for significance (FDR-q = 0.05 and p<0.001; see Materials and methods for details) and leave-one-out probabilistic functional maps highlighting voxels that are significant in at least three of the other nine subjects. For each participant, CN/SOC and IC are shown in transversal cuts, MGB is shown in a coronal cut. See single subject videos for 3-D view of these maps in *Figure 10* supplements. Unthresholded maps can be found in our online resources (see Data Availability section).

*Figure 4 continued on next page*

*Figure 4 continued*

DOI: https://doi.org/10.7554/eLife.48932.009

The following figure supplements are available for figure 4:

**Figure supplement 1.** Correspondence between single subject activation maps and leave-one-out probabilistic maps.

DOI: https://doi.org/10.7554/eLife.48932.010

**Figure supplement 2.** Effect of threshold on leave-one-out probabilistic maps on correspondence with single subject activations].

DOI: https://doi.org/10.7554/eLife.48932.011

**Figure supplement 3.** Reproducibility across experiments of the functional activation maps in six participants (also see *Figure 11*).

DOI: https://doi.org/10.7554/eLife.48932.012

**Figure supplement 4.** Correspondence between single subject activation maps across experiments.

DOI: https://doi.org/10.7554/eLife.48932.013

**Figure supplement 5.** Effect of spatial smoothing in the analysis of the data collected from two of the participants.

DOI: https://doi.org/10.7554/eLife.48932.014

individual participants. For this reason, we collected a large quantity of functional data in all individuals: two sessions with 12 runs each in Experiment 1 and 2 sessions with eight runs each in Experiment 2 (totalling 8 hr of functional data for each participant who completed both experiments). All statistical analyses were performed at the single subject level. Group analyses were used to evaluate the correspondence across subjects of individually defined regions (i.e. the definition of a probabilistic atlas across participants) as well as the ability to generalize to new participants by means of a leave-one-out analysis.

## Anatomical MRI

Visual inspection and comparison to the MNI dataset (*Figure 5—figure supplement 2*) showed that the MGB and IC could be identified on the basis of the anatomical contrast in our participants (*Figure 5—figure supplement 1*), especially in the short inversion time T1-weighted data (*Tourdias et al., 2014*; *Moerel et al., 2015*). However, while the superior olivary complex (SOC) could be identified in the MNI dataset (*Figure 5—figure supplement 2*), it could not be identified in average anatomical image from our 7T data. This is possibly due to the limited number of subjects leading to the lower signal to noise in the average image. We have also explored the combination of image contrasts within each individual using a compositional method proposed in *Gulban et al. (2018b)*, but the results were inconclusive.

## Functional MRI

The difficulty in delineating the CN and SOC from anatomical in vivo MRI data (see *Figure 5—figure supplement 1* for the average anatomical images obtained from our in vivo data) oriented our investigation towards the possibility to identify the subcortical auditory pathway—in vivo and in single individuals—on the basis of the functional responses to sounds. Functional responses to 168 natural sounds (Experiment 1) were collected at 7T using a sparse acquisition scheme and a fast event-related design. We additionally report the reproducibility of the individual functional delineations in six out of the 10 participants who participated in a follow up experiment in which responses to 96 natural sounds (Experiment 2) were collected at 7T using a sparse acquisition scheme and a fast event-related design.

Statistical analysis of the functional responses allowed us to define voxels with significant activation in response to sounds in each individual. Additionally, we created a probabilistic functional atlas based on the overlap of statistically significant maps across individuals (after anatomical registration to a reference subject). *Figure 5* shows the overlap of functional responses across participants, plotted on top of in vivo anatomical MRI, histology, and post mortem MRI. To evaluate the generalization to new data we also computed leave-one-out probabilistic functional atlases each time leaving one one of our participants (see Materials and methods for details).

*Figure 4* shows, for each individual participant, the statistically thresholded (see Materials and methods) activation maps together with leave-one-out probabilistic functional maps obtained considering all other individuals. The unthresholded maps are reported in supplement videos to *Figure 4* and available for inspection in the online repository of the data. In all our

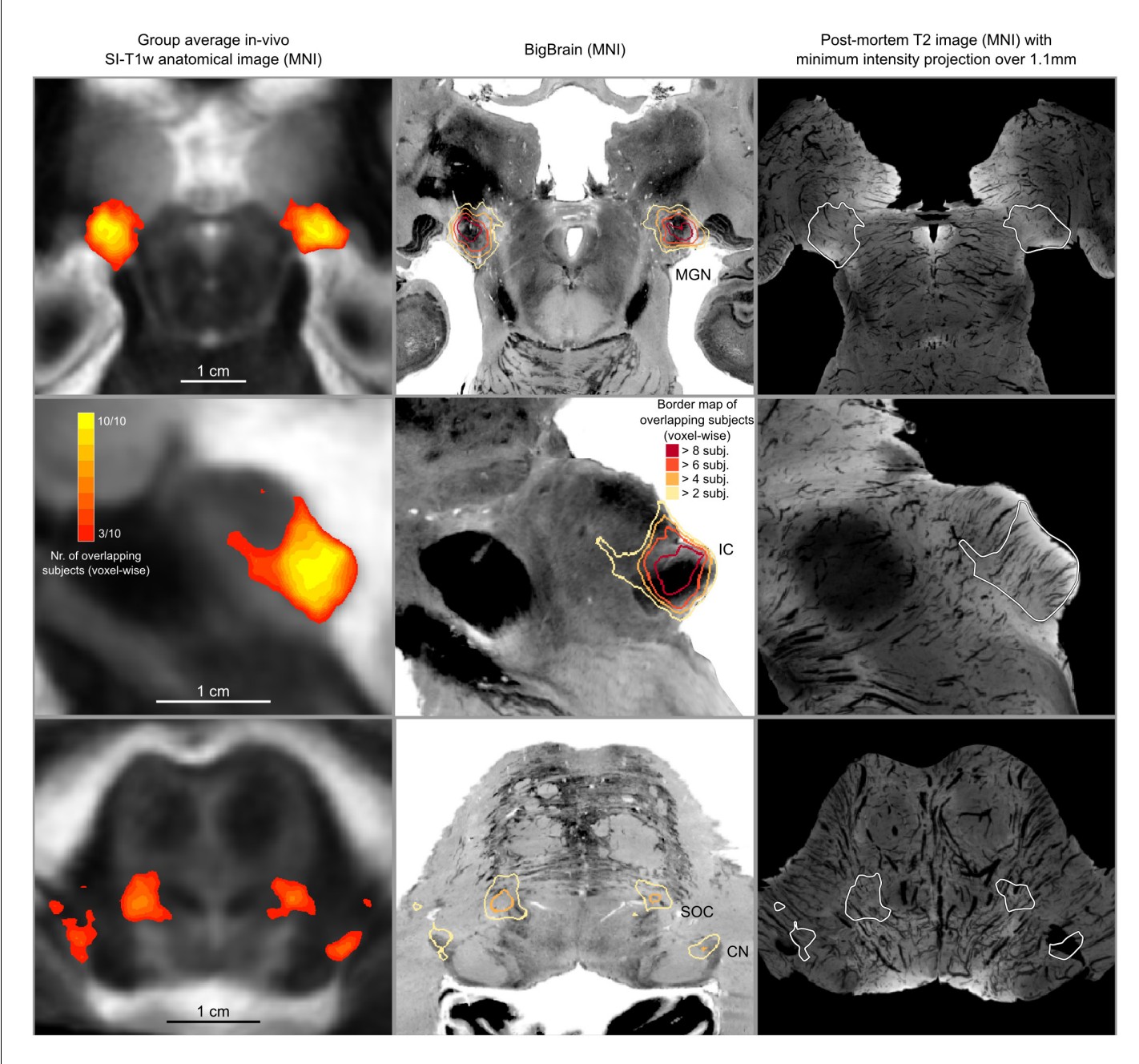

**Figure 5.** In vivo functional MRI responses to auditory stimuli, combined across 10 participants. Left column: Conjunction of participants plotted on top of one participant's short inversion T1-weighted anatomical MRI. Center column: Conjunction of participants' fMRI responses warped to MNI space and plotted on top of BigBrain MNI (corrected) image. Right column: Conjunction of fMRI responses plotted on top of post mortem MRI vasculature images (1.1 mm minimum intensity projection).

DOI: https://doi.org/10.7554/eLife.48932.015

The following figure supplements are available for figure 5:

**Figure supplement 1.** In vivo anatomical group average images in MNI space.

DOI: https://doi.org/10.7554/eLife.48932.016

**Figure supplement 2.** Anatomical images from MNI ICBM 152 compared to BigBrain in MNI space Anatomical images from MNI ICBM 152 2009b dataset compared to BigBrain histology in MNIspace (left column).

DOI: https://doi.org/10.7554/eLife.48932.017

participants, we could identify clusters of significant activation in response to sounds in the MGB, IC, SOC, and CN. In each individual and for each auditory nucleus, these activation clusters correspond to locations that are significantly active in at least three out of the other nine participants to the experiment. *Figure 4—figure supplement 1* reports the overlap and distance between functional centroids of the single subject activation maps and the leave-one-out probabilistic maps. In addition, *Figure 4—figure supplement 3* shows the reproducibility of the functional responses across experiments in six of the participants. The analysis of the overlap and distance between the centroids of activation across experiments within each of these six participants is reported in *Figure 4—figure supplement 4*. The higher signal-to-noise ratio attainable in regions corresponding to the IC and MGB results in highly reproducible functional responses both within and across participants in these regions. Activation clusters identified at the level of CN and SOC in single individuals also reproduce (albeit to a smaller degree with respect to IC and MGB), both within subjects (i.e. across experiments) and across subjects.

The left column of *Figure 5* shows the probabilistic functional map obtained from all participants in Experiment 1 (i.e. representing the number of subjects in which each voxel was identified as significantly responding to sounds-the map is thresholded to display voxels that are significantly activated in at least three out of the 10 participants) overlaid on the in vivo average anatomical MRI image (short inversion time T1-weighted image [*Tourdias et al., 2014*]; see Materials and methods for details).

Projecting these data to the reference MNI space allowed evaluating the correspondence between in vivo functionally defined regions and histological data (Big Brain - *Figure 5*, center column).

At the level of the CN, the clusters of voxels active in at least three out of the 10 participants correspond mostly to the ventral part of CN. The dorsal subdivision of the CN is not recovered in these probabilistic maps (at least not in at least three volunteers consistently) possibly due to partial voluming with the nearby cerebrospinal fluid in combination with thinness (thickness around 0.5 mm) of the dorsal CN as it wraps around inferior cerebellar peduncle (see *Figure 1*). Nearby, the location of the activation clusters identifying the SOC overlaps with the SOC as identified in the BigBrain data.

As the next step, we qualitatively investigated if the orientation of the vasculature at the level of the SOC may have an effect on size (and location) of the functionally defined regions. As a visual aid in this evaluation, we overlaid the functionally defined regions with the vasculature image obtained from the postmortem data (*Figure 5*, right column). In all subcortical regions, the vasculature appears to have a specific orientation, and, at the level of the SOC, vessels drain blood from the center in a ventral direction (i.e. the direction of draining is toward the surface of the brainstem in the top of the image reported in the transverse view, bottom in *Figure 5*). This specific vasculature architecture may result in the displacement or enlargement of the functionally defined clusters toward the ventral surface of the brainstem (as highlighted in the correspondence with histological data in *Figure 5*).

The probability of the same voxel to be significantly modulated by sound presentation across subjects increased at the level of the IC and MGB, where the histologically defined regions corresponded (for the large part) to all subjects exhibiting significant responses to sounds. At the threshold of three subjects in the probabilistic maps, the IC seems to extend toward the superior direction, bordering and sometimes including parts of superior colliculus. On the other hand, similarly to what may happen in the SOC, the general directions of the vasculature penetrating the IC and draining blood towards the dorsal surface of brainstem angled in a superior direction (*Figure 5* right panel) may also impact the functional definition of the IC.

The functional responses in the MGB cover an area that is in agreement with histological data. Interestingly, compared to the IC or SOC, there is no major direction of extension of functional responses as well as no clear direction (in comparison to SOC and IC) of vascular draining.

A quantification of the volume of functionally defined structures is reported in *Table 1* for different thresholds of the probabilistic group map (from a threshold that defines the regions based on voxels that are significant in at least three out of the 10 participants to a threshold that define the regions based on voxels that are significant in at least five out of the ten participants). The overlap between functional regions and the BigBrain segmentations after projection in MNI space is reported in Table 2 (as bottom right inset in *Figure 2* - computed using a threshold for the

probabilistic maps that defines the regions based on voxels that are significant in at least three of the 10 participants).

## Diffusion MRI

With the successful identification of the subcortical auditory structures with functional MRI, we next sought to estimate the likely connections between these structures in vivo. We analyzed the high spatial and angular resolution diffusion data to estimate streamlines of white matter connectivity following a similar process as the postmortem MRI (see Materials and methods for further details).

*Figure 6* shows diffusion tractography streamlines that pass through at least one subcortical auditory structure (as defined by group-level probabilistic functional activation [significant response in at least three out of 10 subjects]; see section above). The high spatial and angular resolution of these

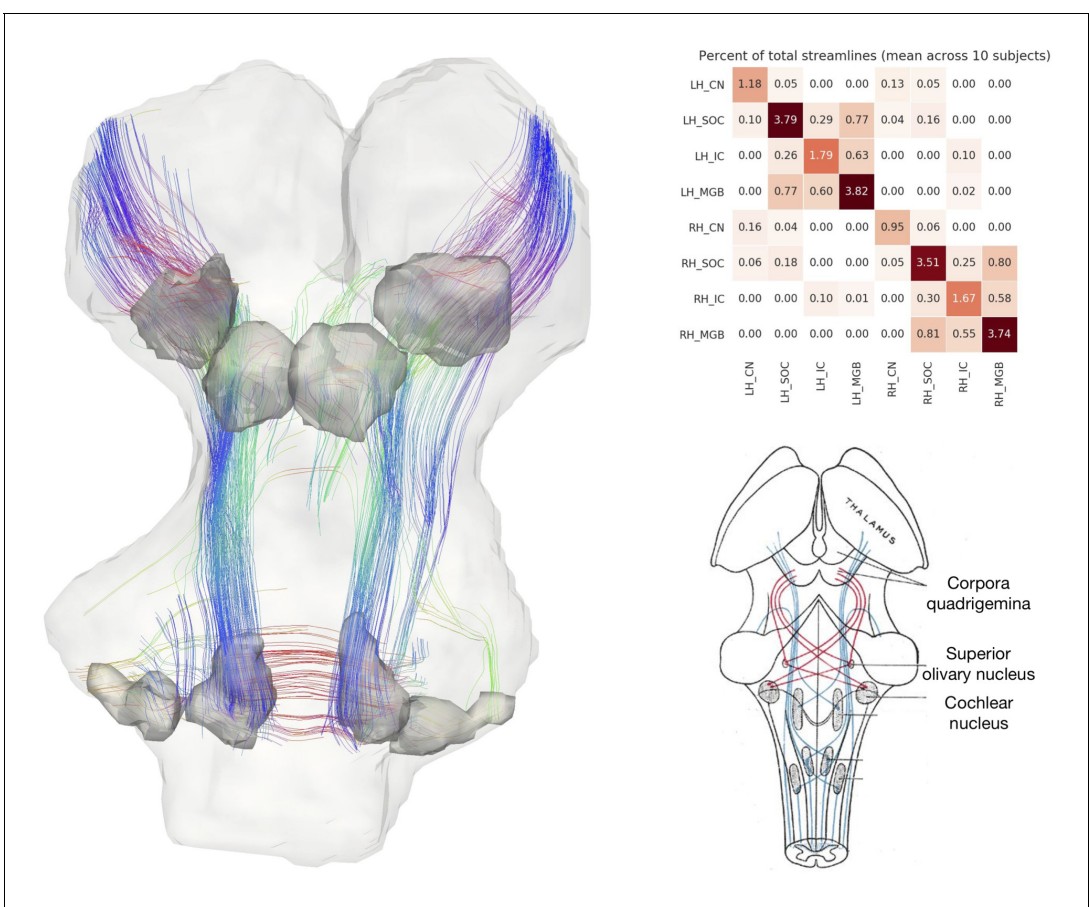

**Figure 6.** In vivo tractography of the subcortical auditory system from 7T diffusion-weighted MRI. Left: 3-D images from one participant. Fiber orientation distribution functions were estimated from diffusion-weighted MRI images of the brainstem and were used for deterministic tractography. Streamlines that passed through functionally defined auditory ROIs (dark grey) are shown here (excluding streamlines through the medulla). Colors represent the local orientation at each specific point along the streamline: blue is inferior-superior, red green is anterior-posterior, and red is left-right. A rotating animation is available in the online resources. Top right: connectivity between subcortical auditory ROIs as a percentage of total brainstem streamlines, averaged over 10 participants. Bottom right: schematic of auditory brainstem connectivity from Gray's Anatomy of the Human Body. A video of the streamlines is available online: https://osf.io/ykd24/.
DOI: https://doi.org/10.7554/eLife.48932.018

The following video and figure supplement are available for figure 6:

**Figure supplement 1.** Bar plot of streamline counts through each ROI.
DOI: https://doi.org/10.7554/eLife.48932.019
**Figure 6— video 1.** 360˚ rotation video of in vivo streamlines.
DOI: https://doi.org/10.7554/eLife.48932.020

data allow for vastly improved estimation of white matter connections between these deep, small structures.

While not a measure of actual physical brain connections—and therefore requiring caution in interpretation—connectivity patterns resemble what we would expect to see based on animal model tracer investigations. Overall, the connectivity network appears to be dominated by laterality, in that left hemisphere structures are generally more connected with other left hemisphere structures.

However, there are a few notable exceptions to this pattern: the cochlear nuclei and superior olivary complexes are strongly connected bilaterally, which fits with animal research suggesting one-half to two-thirds of ascending auditory connections cross the midline at these early stages. Additionally, there are a small number of connections between left and right inferior colliculi, likely along the anatomical commissure of the inferior colliculus.

## Discussion

The auditory pathway includes a number of subcortical structures and connections, but identifying these components in humans has been challenging with existing in vivo imaging methods. We showed that functional localization of the subcortical auditory system is achievable within each participant, and that localization is consistent across experimental sessions. To further facilitate research on the anatomy and function of the human subcortical auditory system, we created 3-D atlases of the human auditory pathway based on gold standard histology, 50 μm isotropic resolution post mortem anatomical MRI, and in vivo functional MRI at 7T. In addition, we created 3-D connectivity maps of the human subcortical auditory pathway using diffusion MRI tractography in a postmortem MRI sample and in living participants.

These atlases and connectivity maps are the first fully 3-D representations of the human subcortical auditory pathway and are publicly available to make the localization of subcortical auditory nuclei easier. In particular, the atlases are available in a common reference space (MNI152) to make registration to other MRI data as straightforward as possible. As part of this registration process, we have improved the registration of the brainstem of BigBrain histological data to the MNI space, where the original MNI version presented a significant misregistration of the colliculi (as noticeable in *Figure 7*). The result of our new registration allows to more correctly localize the colliculi of BigBrain data in MNI without compromising the registration of other brainstem and thalamic nuclei.

In creating the atlas with three distinct modalities, we were able to assess the reliability of each of the methods in identifying the human subcortical auditory pathway. Each modality provided useful information to the segmentation of the auditory nuclei. All regions could be identified in the Big-Brain histological data, that also allowed us to identify small auditory sub-nuclei such as the medial superior olive and lateral superior olive. High-resolution post mortem MRI also clearly delineated the medial geniculate and inferior colliculus (with less contrast for the superior olive and cochlear nucleus), while the overall image contrast facilitated registration with in vivo MRI. High-resolution in vivo functional MRI exhibited greater sensitivity to auditory structures than in vivo anatomical MRI that was even higher resolution. We showed that functional MRI is useful to localize structures throughout the auditory pathway despite their small size. In each participant, we identified voxels significantly responding to sound presentation in regions corresponding to the CN, SOC, IC and MGB. We validated these definition by evaluating both the within-subject reproducibility (i.e., by comparing functional maps across two experiments in six individuals) and the ability of a probabilistic atlas defined on nine out of our 10 participants to generalize to the left out volunteer.

In total, we found that each of the methods described here provides information to the delineation of the human subcortical auditory pathway. Our post mortem and in vivo data suggest that MRI is a capable tool for investigating this system across spatial scales providing a bridge to the gold standard, histology.

While not representing specific cells, MRI holds a number of advantages over the gold standard method, histology (*Johnson et al., 1993*). First, MRI allows for visualization and analysis of an entire 3-D structure at once, with minimal geometric warping from (virtual) slice to slice (which can occur in slice-based histology if individual slices contract on a slide or are damaged during the physical slicing). Second, MRI can be used in vivo in human participants, opening up the possibility to address research questions on the functional and anatomical properties of human subcortical structures, their correspondence, and their involvement in human behavior.

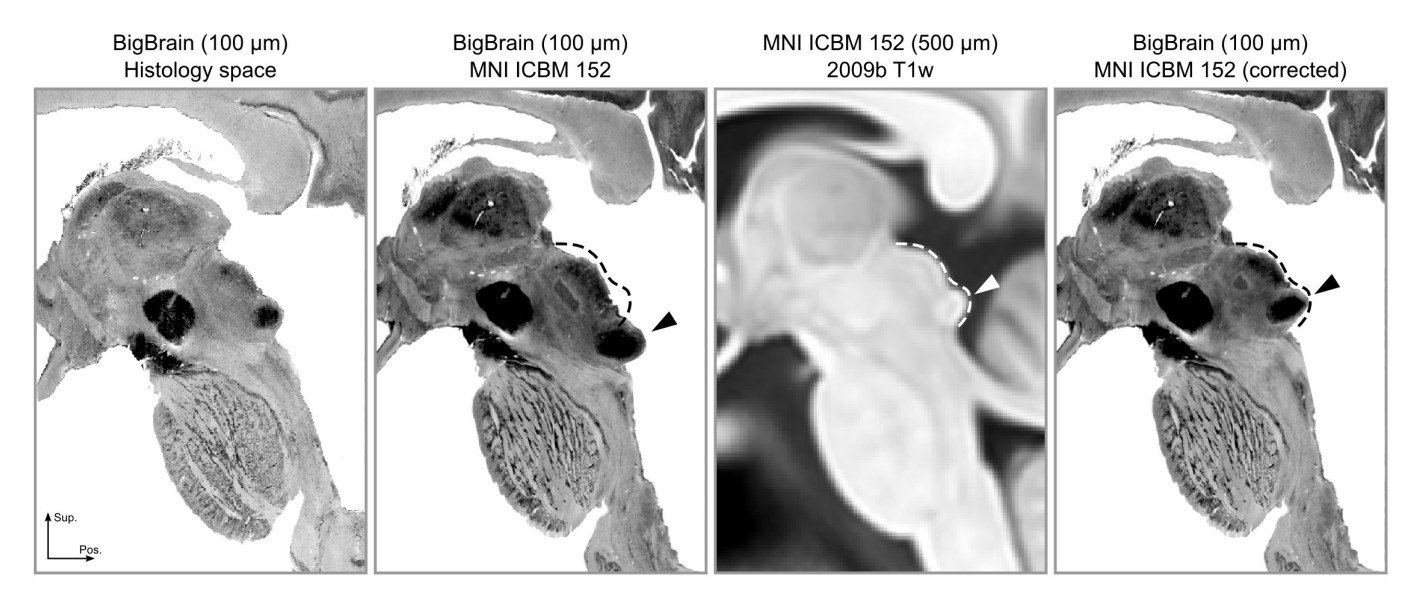

**Figure 7.** The registration error around the inferior colliculus is visible bilaterally when comparing Panel 2 and Panel 3. The dashed lines indicate the correct shape (and location) of the colliculi in MNI space. The arrows point to the inferior colliculus (IC). The last panel shows the corrected BigBrain MNI dataset.

DOI: https://doi.org/10.7554/eLife.48932.021

Probing the connectivity of the human subcortical auditory pathway has been extremely limited, since gold standard (but invasive) tracer studies are largely unavailable for human specimens. In this study, we show that diffusion MRI tractography is sensitive to connections within the human subcortical auditory system, both post mortem and in vivo. In addition to streamlines corresponding to the lateral lemniscus-the major ascending auditory white matter tract-we can see streamlines crossing the midline at the level of the superior olivary complex and the inferior colliculus.

Interestingly, with the highest resolution data (200 µm postmortem diffusion-weighted MRI), we were able to estimate streamlines visually resembling the expected auditory pathway, but missing putative key connections between subcortical auditory structures themselves when using the strictly defined ROIs as tractography seeds. In contrast, the relatively lower resolution in vivo diffusion-weighted MRI produced estimates of connectivity more like what we expected from the literature. We had two hypotheses as to why these results appeared. First, the higher resolution anatomical definition of the nuclei not including the immediately surrounding white matter could miss streamlines that terminate at the immediate proximity of the structures' borders (similar to issues in cortex *Reveley et al., 2015*). Second, partial volume effects in the lower resolution data—combining white matter and grey matter in the same voxels—could actually *increase* streamlines terminating within the anatomical ROIs. Dilating the post mortem ROIs and downsampling the data to the in vivo resolution both resulted in greater streamline connectivity between subcortical auditory structures, suggesting that our hypotheses were likely. Thus, while high spatial resolution diffusion-weighted MRI allowed for much finer, higher quality streamline estimates, it also places constraints on tractography analyses that must be accounted for and investigated further.

More generally, the density of brainstem and midbrain nuclei and frequent crossings between perpendicular white matter bundles pose a challenge to diffusion tractography estimations of white matter connectivity, so it was not clear beforehand if this methodology would be sufficient for visualizing these connections. Additionally, because a gold-standard connectivity method is not available in humans, we could not directly validate our tractography findings (as can be done in the macaque, though with limited success; see *Thomas et al., 2014*). However, our results suggest that, with continually improving diffusion-weighted MRI acquisition and analysis techniques, focused investigations on the human subcortical auditory pathway can-and should-become more prominent in the near future.

In addition to high-resolution anatomical postmortem MRI and diffusion MRI tractography, we were also able to identify the subcortical auditory system in vivo with functional MRI. Previous studies have identified these structures with functional MRI, but they typically required constrained acquisition parameters—for instance, they used single slices with low through-plane resolution in order to support high in-plane resolution (*Guimaraes et al., 1998*; *Harms and Melcher, 2002*; *Griffiths et al., 2001*; *Hawley et al., 2005*; *Sigalovsky and Melcher, 2006*). In the present study, by taking advantage of the increased signal of high-field (7-Tesla) MRI, we were able to image the brainstem using isotropic voxels at high resolution across a wider field-of-view that covers the human auditory pathway in coronal oblique slices. The use of slice acceleration (*Moeller et al., 2010*; *Setsompop et al., 2012*) allowed us to acquire enough slices to cover the whole brainstem, thalamus and cortical regions around Heschl's gyrus with the exclusion of anterior portions of the superior temporal gyrus and sulcus. Using isotropic voxels allowed us to better evaluate the 3-D volume of significantly activated regions, limiting partial volume effects that are inevitable when using thick anisotropic slices.

Similar to previous research at lower magnetic fields (*Hawley et al., 2005*; *Sigalovsky and Melcher, 2006*), the 7T MR images did not allow for an anatomical definition of the CN and SOC (although IC and MGB were clearly visible). A possible reason for this is the reduced signal- and contrast-to-noise ratio in these regions. Only very recently has 7T MRI enabled anatomical localization at the level of the SOC in individual subjects (*García-Gomar et al., 2019*). It should be noted that we could identify the SOC in the MNI ICBM 152 dataset that results from the average of a much larger cohort. Therefore, future investigations should be tailored to optimize anatomical image contrasts to auditory brainstem regions in single subjects. The (postmortem) atlases we provide here will prove a useful tool for these investigations by providing a reference for the expected location (and size) of these regions.

In contrast to in vivo anatomical localization, our data—in agreement with previous reports (*Hawley et al., 2005*; *Sigalovsky and Melcher, 2006*)—show that *functional* mapping of the subcortical auditory pathway is an effective method for localizing these structures. While histologically defined CN and SOC regions have been previously used to sample functional responses from in vivo fMRI data (*Hawley et al., 2005*; *Sigalovsky and Melcher, 2006*), the overlap between functionally and histologically defined subcortical auditory structures has not been reported before. Here, we investigated the ability of BOLD fMRI (as an indirect measure of neuronal activity) to localize subcortical auditory regions. We show that functional definitions are possible, as distinct clusters of activation were detected in all subjects across the subcortical auditory pathway. These regions were reproducible both within subjects (across experiments) and across subjects (comparing single participants functional maps to the leave-one-out atlas obtained with all other participants). We could identify the subcortical auditory nuclei despite not using cardiac gating, a method that previous studies showed to increase the signal-to-noise ratio in subcortical regions (*Guimaraes et al., 1998*; *Harms and Melcher, 2002*; *Griffiths et al., 2001*; *Hawley et al., 2005*; *Sigalovsky and Melcher, 2006*). We instead increased statistical power by presenting a large number of natural sounds with multiple repetitions. Using smaller voxels also reduced partial volume effects between cerebrospinal fluid (which is heavily affected by physiological noise) and the brain tissue (*Triantafyllou et al., 2016*). In addition, the correspondence of functionally defined regions across ten participants after anatomical alignment allowed us to build a functional probabilistic atlas.

Despite these positive outcomes, functionally defined regions exhibited overall larger volumes compared to the histological ones (see *Table 1* in *Table 1*). Although we acquired data at relatively high resolution (1.1 mm isotropic), our functional voxel size and the mild spatial smoothing (1.5 mm) might be the source of this observation. Another factor that may have impacted the increased volume of the in vivo probabilistic regions can be the residual anatomical misalignment across subjects that also contributes especially to the lower degree of overlap at CN and SOC. In this case, the individual anatomical images not showing enough contrast might be the cause. Partial volume also most likely impacted small regions such as the CN and SOC, and draining effects due to the vascular architecture could also have an impact on the size and localization of the in vivo defined regions. Further, because we used only the overall response to sounds as functional definition, the regions we defined may include sub-regions not specific to the system under investigation (e.g. the inclusion of multisensory deep layers of the superior colliculus at the border with the IC; *Sparks and Hartwich-Young, 1989*; *Jiang et al., 1997*). This effect could be reduced by using different stimuli and

statistical contrasts. For instance, one could contrast uni-sensory and multi-sensory stimuli to identify—within the current functional definition—the IC voxels that respond to visual stimulation and thus may represent multi-sensory superior colliculus. For the IC and MGB, where signal-to-noise ratio in the functional data is larger, a higher threshold in the probabilistic maps results in a more accurate volumetric definition as well as more correct anatomical localization (see, e.g. *Figure 5*). It should also be noted that direct comparison of post-mortem and in vivo results suffers from the additional problem of aligning data with very diverse contrasts and resolutions. For the IC and MGB, our procedure could be verified on the basis of the anatomical contrast in the in vivo data, for the CN and SOC the lack of anatomical contrast (to be leveraged by the alignment procedure) in the in vivo data may be the source of some of the misalignment between the data.

We also investigated the possibility of defining anatomical connections between subcortical auditory nuclei using diffusion-weighted MRI. While affected by similar confounds as functional MRI (e.g. partial voluming effects, physiological noise, and relative signal weighting), this technique faces additional complications introduced by the number of orientations required, the gradient strength (b-value) selected, the modeling of diffusion or fiber orientations within each voxel, and the estimation of streamlines across brain regions, especially within the subcortical auditory system (*Zanin et al., 2019*). The post mortem and in vivo diffusion MRI datasets in this study each implemented state-of-the-art acquisition techniques to optimize the MRI signal-to-noise ratio and minimize MRI modeling errors. For example, as the fixation process likely changes the diffusion characteristics of the tissue (*Pfefferbaum et al., 2004*; *Miller et al., 2011*), we compensated for this effect by increasing the diffusion gradient strength (b-value). The constrained spherical deconvolution modeling method takes advantage of the high angular resolution of each dataset to provide fine-grained estimations of fiber orientation distributions. Additionally, the Euler Delta Crossings (EuDX) deterministic tractography method is effective at generating streamlines through voxels with multiple fiber orientation peaks, such as where white matter bundles cross. However, as diffusion MRI and tractography are not measuring true neuronal connections, there is still room for error in diffusion orientation and streamline estimation (*Schilling et al., 2019a*; *Schilling et al., 2019b*).

Our BigBrain histological segmentations are very similar in volume to those reported previously in the literature (*Moore, 1987*; *Glendenning and Masterton, 1998*), with slightly smaller cochlear nuclei and slightly larger medial geniculate bodies, but similar SOC and IC volumes. It has to be noted that the physical slicing process potentially introduces deformations in the tissue, and while the publicly available BigBrain dataset is of extremely high quality (with good registration from slice to slice), subtle deformations may have affected the shape or volume of the structures we identified.

Postmortem MRI segmentations differed more greatly, with smaller CN and SOC definitions but larger MGB definitions compared to both the literature and BigBrain histological segmentations. These differences could possibly be caused by the reduced contrast-to-noise ratio in the post mortem MRI data compared to the histological data (despite their high spatial resolution). This reduced contrast-to-noise ratio may be caused by both reduced differences in magnetic properties between the regions and their surrounding tissues as well as from residual partial volume effects (especially for the very small sections of the dorsal CN, for example) that may have blurred the borders of the auditory nuclei in the post mortem MRI data. Contrast-to-noise ratio may be ameliorated by different acquisition/reconstruction techniques (*Wang et al., 2018*), and optimizing parameters may improve the definition of auditory nuclei on the basis of postmortem MRI data. Finally, slight misregistration between specimens (e.g. the histological data and the postmortem MRI data) likely still affect our comparisons, as registration between images (particularly from different modalities) remains a challenge. For instance, *Figure 2* shows slightly different shapes and locations for the inferior colliculus between the two datasets, despite non-linear registration to the same template. Although non-linear methods significantly improve gross registration between specimens, large misregistrations are still possible (as shown for the colliculi in the original BigBrain MNI registration). These issues can be addressed manually using additional image registration techniques, as we did here with the BigBrain MNI registration (see our 'corrected' version above), but such hands-on, time-intensive edits are not always possible. Further, vastly different image contrasts (like histology and MRI) result in different regions or subregions being emphasized in the signal, creating an additional challenges in the registration procedure.

More generally, post mortem imaging—whether MRI or histology—is prone to modest deformation of the specimen. Additionally, both post mortem specimens in this paper (BigBrain and post

mortem MRI) were from 65-year-old male donors, and age may have additionally affected the volume of the brain structures we investigated.

Despite these limitations, the inter-rater and inter-experiment reliability in this study suggest that each method is effective for localizing the subcortical auditory pathway. The reliable functional localization of subcortical auditory structures opens the door to future investigations of more complex human auditory processing. The atlases derived from each localization method is publicly available (see 'Data and code availability' in Materials and methods) to facilitate further investigations into the structure, function, and connectivity of the human subcortical auditory system in vivo. Lastly, the 3-D representations found in this paper and in the available data should be beneficial to others in understanding the immensely complex, but identifiable, structure of the human subcortical auditory pathway.

## Materials and methods

See *Figure 8* for a summary of data sources, data processing steps, and software used in these analyses.

### MRI acquisition parameters

#### In vivo MRI

The experimental procedures were approved by the ethics committee of the Faculty for Psychology and Neuroscience at Maastricht University (reference number: ERCPN-167_09_05_2016), and were performed in accordance with the approved guidelines and the Declaration of Helsinki. Written informed consent was obtained for every participant before conducting the experiments. All participants reported to have normal hearing, had no history of hearing disorder/impairments or neurological disease.

Images were acquired on a 7T Siemens MAGNETOM scanner (Siemens Medical Solutions, Erlangen, Germany), with 70 mT/m gradients and a head RF coil (Nova Medical, Wilmington, MA, USA; single transmit, 32 receive channels) at Maastricht University, Maastricht, Netherlands.

We conducted two separate experiments. In Experiment 1, data were collected for $n$=10 participants (age range 25 to 30, six females), in three separate sessions. In the first session, we acquired the in vivo anatomical data set consisting of: 1) a T1-weighted (T1w) image acquired using a 3-D MPRAGE sequence (repetition time [TR]=3100 ms; time to inversion [TI]=1500 ms [adiabatic non-selective inversion pulse]; echo time [TE]=2.42 ms; flip angle = 5°; generalized auto-calibrating partially parallel acquisitions [GRAPPA]=3 (*Griswold et al., 2002*); field of view [FOV]=224 × 224 mm$^2$; matrix size = 320 × 320; 256 slices; 0.7 mm isotropic voxels; pixel bandwidth = 182 Hz/pixel; first phase encode direction anterior to posterior; second phase encode direction superior to inferior); 2) a Proton Density weighted (PDw) image (0.7 mm iso.) with the same 3-D MPRAGE as for the T1w image but without the inversion pulse (TR = 1380 ms; TE = 2.42 ms; flip angle = 5°; GRAPPA = 3; FOV = 224 × 224 mm$^2$; matrix size = 320 × 320; 256 slices; 0.7 mm iso. voxels; pixel bandwidth = 182 Hz/pixel; first phase encode direction anterior to posterior; second phase encode direction superior to inferior); 3) a T2*-weighted (T2w) anatomical image acquired using a modified 3-D MPRAGE sequence (*De Martino et al., 2015*) that allows freely setting the TE (TR = 4910 ms; TE = 16 ms; flip angle = 5°; GRAPPA = 3; FOV = 224 × 224 mm$^2$; matrix size = 320 × 320; 256 slices; 0.7 mm iso. voxels; pixel bandwidth = 473 Hz/pixel; first phase encode direction anterior to posterior; second phase encode superior to inferior) and 4) a T1-weighted images acquired with a short inversion time (SI-T1w) using a 3-D MPRAGE (*Tourdias et al., 2014*) (TR = 4500 ms; TI = 670 ms [adiabatic non-selective inversion pulse]; TE = 3.37 ms; flip angle = 4°; GRAPPA = 3; FOV = 224 × 224 mm$^2$; matrix size = 320 × 320; 256 slices; 0.7 mm isotropic voxels; pixel bandwidth = 178 Hz/pixel; first phase encode direction anterior to posterior; second phase encode direction superior to inferior). To improve transmit efficiency in temporal areas when acquiring these anatomical images we used dielectric pads (*Teeuwisse et al., 2012*).

In the same session we acquired, for each participant, a diffusion-weighted MRI data set using a multi-band diffusion-weighted spin-echo EPI protocol originating from the 7T Human Connectome Project (1.05 mm isotropic acquisition and b-values = 1000 and 2000 s/mm$^2$) (*Vu et al., 2015*), extended in order to collect one additional shell at b-value at b = 3000 s/mm$^2$(*Gulban et al., 2018a*). Other relevant imaging parameters were (FOV = 200 × 200 mm$^2$ with partial Fourier 6/8,

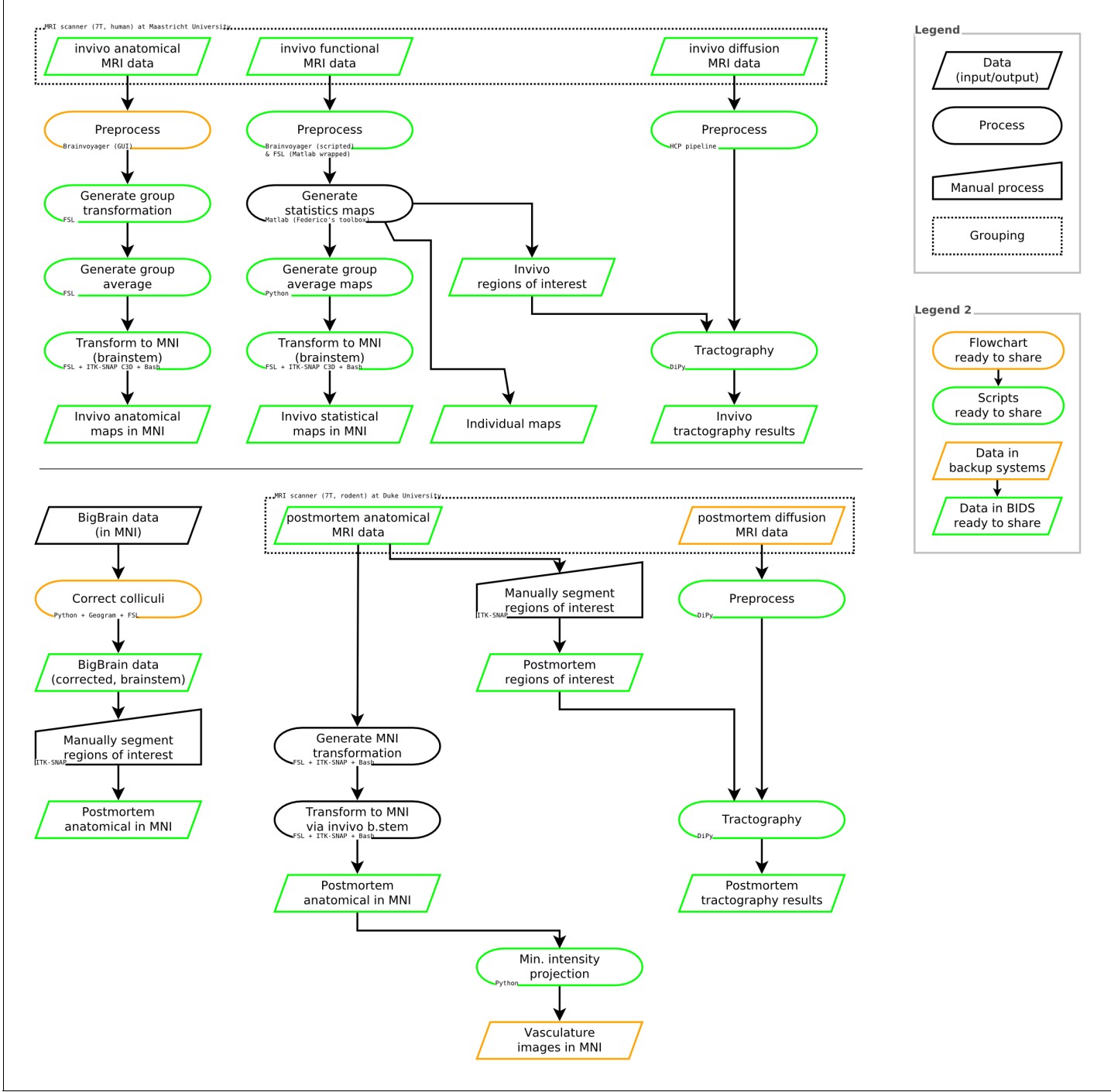

**Figure 8.** Summary of data processing steps, including availability of data and code.

DOI: https://doi.org/10.7554/eLife.48932.022

132 slices, nominal voxel size = 1.05 mm isotropic, TR/TE = 7080/75.6 ms, MB = 2, phase encoding acceleration (GRAPPA) = 3, 66 directions and 11 additional b = 0 volumes for every b-value). A total of 462 volumes were obtained (231 in each phase encoding direction anterior-posterior and posterior-anterior) for a total acquisition time of 60 min.

The other two sessions were used to collect functional data in order to identify sound responsive regions in the human thalamus and brainstem. Participants listened to 168 natural sounds (1 s long)

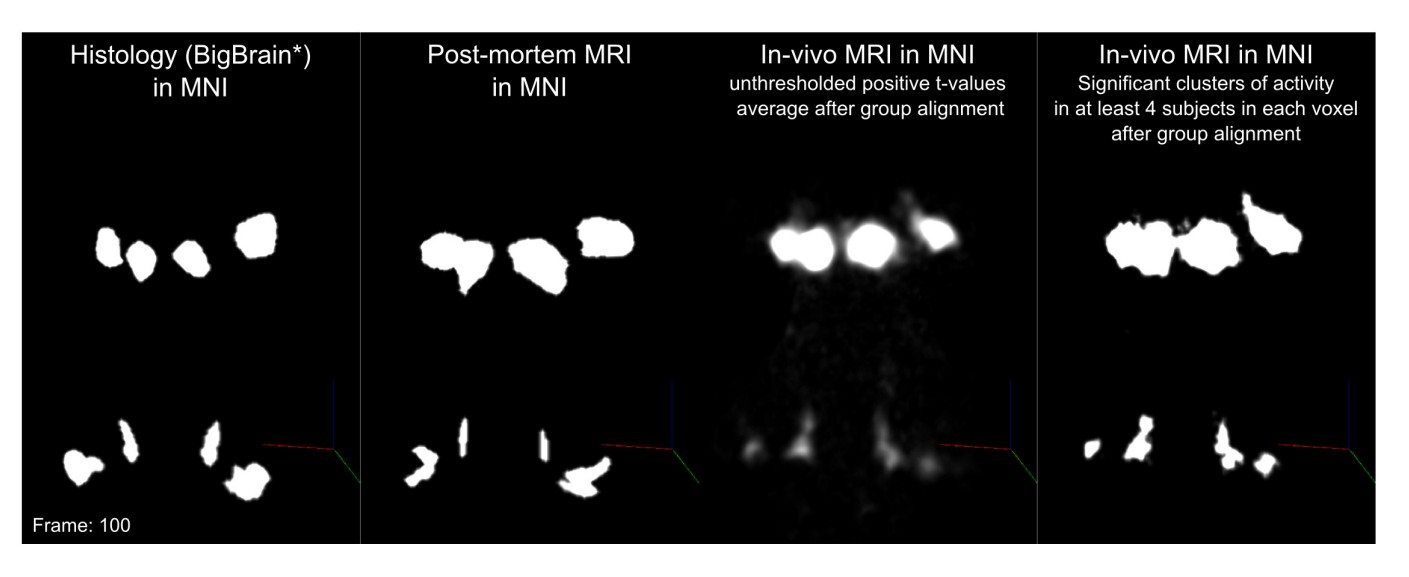

**Figure 9.** One frame of volume rendered animations for comparing histology (BigBrain), post-mortem MRI, in-vivo MRI unthresholded positive t-values group average and in-vivo MRI clusters of significant activity overlapping in at least four subjects in each voxel.

DOI: https://doi.org/10.7554/eLife.48932.023

The following video is available for figure 9:

**Figure 9— video 1.** 3D volume rendered comparisons in MNI space.

DOI: https://doi.org/10.7554/eLife.48932.024

coming from seven categories (speech, voice, nature, tools, music, animals and monkey calls) presented in silent gaps in between the acquisition of functional volumes and were asked to press a button every time the same sound was repeated. The experimental paradigm followed a rapid-event-related design in which sounds were presented with a mean inter stimulus interval of four volumes (minimum three maximum five volumes). The two sessions were identical and each session consisted of twelve functional runs and across the 12 runs each sound was presented three times (i.e. each sounds was presented six times across the two sessions). The 168 sounds were divided in four sets of 42 sounds, each set was presented in three (non consecutive) runs. As a result, the 12 functional runs of each session formed four cross validation sets each one consisting of nine training runs and three testing runs (i.e. 126 training and 42 testing sounds). Note that the testing runs were non overlapping across the cross validations. Catch trials (i.e. sound repetitions) were added to each run, and were excluded from all analyses.

Functional MRI data were acquired with a 2-D Multi-Band Echo Planar Imaging (2D-MBEPI) sequence (*Moeller et al., 2010*; *Setsompop et al., 2012*) with slices prescribed in a coronal oblique orientation in order to cover the entire brainstem and thalamus and covering primary and secondary cortical regions (TR = 2600 ms; Gap = 1400 ms ; TE = 20 ms; flip angle = 80°; GRAPPA = 3; Multi-Band factor = 2; FOV = 206 × 206 mm$^2$; matrix size = 188 × 188; 46 slices; 1.1 mm isotropic voxels; phase encode direction inferior to superior). Reverses phase encode polarity acquisitions were used for distortion correction. Respiration and cardiac information were collected during acquisition using a respiration belt and pulse oximeter respectively.

In experiment 2, six of the volunteers that participated in experiment one were recalled and functional data were acquired with the same slice prescription and functional MRI parameters as in experiment 1 (2D-MBEPI; TR = 2600 ms; Gap = 1400 ms ; TE = 20 ms; flip angle = 80°; GRAPPA = 3; Multi-Band factor = 2; FOV = 206 × 206 mm$^2$; matrix size = 188 × 188; 46 slices; 1.1 mm isotropic voxels; phase encode direction inferior to superior). Experiment 2 consisted of two sessions in which participants listened to 96 natural sounds (1 s long) coming from six categories (speech, voice, nature, tools, music, animals) together with ripples (bandwidth = 1 octave; center frequency = [300

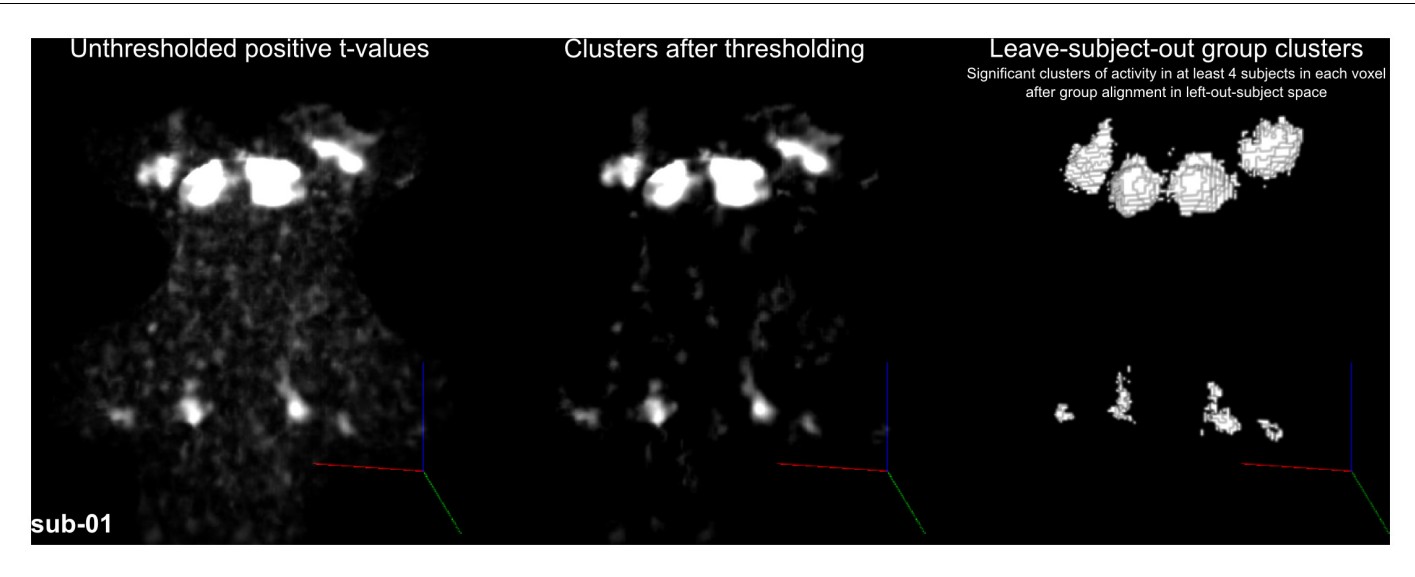

**Figure 10.** One frame of volume rendered animations for single subject statistical maps. (Left)positive t-values (middle) after thresholding (right) leave-one-out probabilistic map ($\geq$ 4)). Viewing angle here is similar to *Figure 1*.

DOI: https://doi.org/10.7554/eLife.48932.025

The following videos are available for figure 10:

**Figure 10—video 1.** Subject 01.

DOI: https://doi.org/10.7554/eLife.48932.026

**Figure 10—video 2.** Subject 02.

DOI: https://doi.org/10.7554/eLife.48932.027

**Figure 10—video 3.** Subject 03.

DOI: https://doi.org/10.7554/eLife.48932.028

**Figure 10—video 4.** Subject 05.

DOI: https://doi.org/10.7554/eLife.48932.029

**Figure 10—video 5.** Subject 06.

DOI: https://doi.org/10.7554/eLife.48932.030

**Figure 10—video 6.** Subject 07.

DOI: https://doi.org/10.7554/eLife.48932.031

**Figure 10—video 7.** Subject 08.

DOI: https://doi.org/10.7554/eLife.48932.032

**Figure 10—video 8.** Subject 09.

DOI: https://doi.org/10.7554/eLife.48932.033

**Figure 10—video 9.** Subject 10.

DOI: https://doi.org/10.7554/eLife.48932.034

**Figure 10—video 10.** Subject 11.

DOI: https://doi.org/10.7554/eLife.48932.035

Hz, 4 kHz]; AM rate = [3 Hz, 10 Hz]). Some ripple sounds contain a short noise burst ('target') and participants were asked to detect such target in either low-frequency ripples or high-frequency ripples in the two sessions respectively (the target occurrence varied (70 vs. 30 percent) for ripples whose center frequency did or did not match the current attention condition). All sounds were presented in silent gaps in between the acquisition of functional volumes. The experimental paradigm followed a rapid-event-related design in which sounds were presented with a mean inter stimulus interval of four volumes (minimum three maximum five volumes). The two sessions consisted of eight functional runs and across the eight runs each natural sound was presented three times (i.e. each sounds was presented six times across the two sessions) while the ripples were presented seven times per run. The 96 natural sounds were divided in four sets of 24 sounds, each set was presented

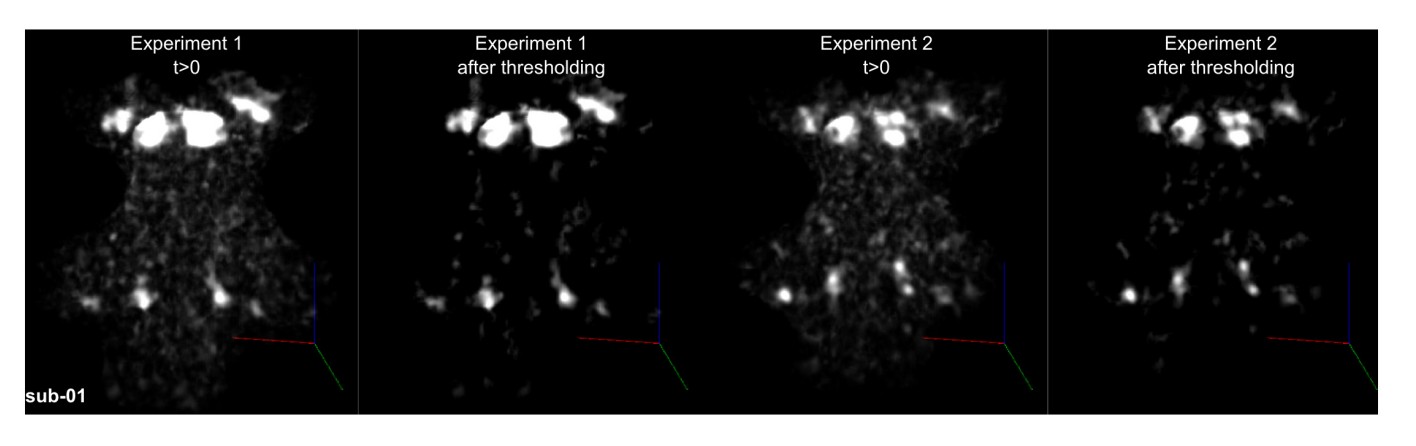

**Figure 11.** One frame of volume rendered animations for Subject 01 statistical maps (experiment 1 positive t-values and thresholded (col 1–2) and experiment 2 positive t-values and thresholded (col 3–4)). Viewing angle here is similar to *Figure 1*.

DOI: https://doi.org/10.7554/eLife.48932.036

The following videos are available for figure 11:

**Figure 11— video 1.** Subject 01 experiment 1 vs experiment 2.

DOI: https://doi.org/10.7554/eLife.48932.037

**Figure 11—video 2.** Subject 02 experiment 1 vs experiment 2.

DOI: https://doi.org/10.7554/eLife.48932.038

**Figure 11—video 3.** Subject 05 experiment 1 vs experiment 2.

DOI: https://doi.org/10.7554/eLife.48932.039

**Figure 11—video 4.** Subject 09 experiment 1 vs experiment 2.

DOI: https://doi.org/10.7554/eLife.48932.040

**Figure 11—video 5.** Subject 10 experiment 1 vs experiment 2.

DOI: https://doi.org/10.7554/eLife.48932.041

**Figure 11—video 6.** Subject 11 experiment 1 vs experiment 2.

DOI: https://doi.org/10.7554/eLife.48932.042

**Figure 11—video 7.** Group average (N=6) unthresholded positive t-values for experiment 1 vs experiment 2.

DOI: https://doi.org/10.7554/eLife.48932.043

in two (non consecutive) runs. As a result, the eight functional runs of each session formed four cross validation sets each one consisting of six training runs and two testing runs (i.e. 72 training natural sounds and 24 testing natural sounds). Note that the testing runs were non overlapping across the cross validations. In each session of experiment 2, we also collected a lower resolution (1 mm isotropic) anatomical reference images (T1 and PD weighted) using the 3D MPRAGE sequence for alignment purposes and included reverses phase encode polarity acquisitions for distortion correction. Respiration and cardiac information were collected during acquisition using a respiration belt and pulse oximeter respectively.

Both in vivo datasets acquired for experiment 1 and experiment 2 have never been published before. This is the first work that uses this dataset.

## Postmortem MRI

A human brainstem and thalamus specimen were dissected at autopsy from a 65-year-old anonymous male. The specimen was flushed with saline and immersed for 2 weeks in 10% solution of neutral buffered formalin. Following this, the specimen was re-hydrated for 1 week in 0.1 M solution of phosphate buffered saline doped with 1% (5 mM) gadoteridol. Before the MRI acquisition, the specimen was placed in custom MRI-compatible tube immersed in liquid fluorocarbon.

Magnetic resonance imaging was conducted in a 210 mm small-bore Magnex/Agilent MRI at the Duke University Center for In Vivo Microscopy. 3-D gradient echo images were collected at 50 $\mu m^3$

spatial resolution over a period of 14 hr, with FOV = 80 × 55 × 45 mm, repetition time (TR) = 50 ms, echo time (TE) = 10 ms, flip angle = 60°, and bandwidth = 78 Hz/pixel.

Diffusion-weighted spin echo images were collected at 200 µm³ spatial resolution with 120 diffusion gradient directions at strength b = 4000 s/m² and 11 b = 0 s/m² volumes over 208 hr. The FOV was 90 × 55 × 45 mm with TR = 100 ms, TE = 33.6 ms, and bandwidth = 278 Hz/pixel.

## Anatomical image registration

SI-T1w, T1w, T2*w and PDw images (700 µm iso.) were transformed to Talairach space (500 µm iso.) using BrainvoyagerQX version 2.8.4 (*Goebel, 2012*). Intensity inhomogeneity correction as implemented in SPM12 unified segmentation (*Ashburner and Friston, 2005*) was used for all images. A smaller volume containing brainstem and thalamus in each image was extracted (in the Talairach space) using FSL version 5.0.9 (*Jenkinson et al., 2012*) and histogram matched using percentile clipping (1% and 99%).

Individual masks for each 10 brainstems were created semi-automatically using ITK-SNAP version 3.6.0 active contour segmentation mode followed by manual edits. These masks included regions starting from 2 cm below the inferior part of pons to 0.5 cm above the medial geniculate nucleus (MGN), with a lateral extend reaching until the lateral geniculate nucleus (LGN) and 3 cm anterior from MGN, not including cerebellum or large arteries that lie on the surface of brainstem. These brainstem masks were then used with FSL-FNIRT (*Andersson et al., 2007*) to warp nine of the 10 brainstems to the reference brainstem (subject 1) using SI-T1w images. We used the SI-T1w images to drive the non linear registration due to the enhanced anatomical contrast across structures within the thalamus and brainstem present in these images (*Tourdias et al., 2014*; *Moerel et al., 2015*). The FNIRT parameters were $\mathrm{subsamp} = 2, 2, 1, 1$, $\mathrm{miter} = 100, 100, 50, 50$, $\mathrm{infwhm} = 2, 2, 1, 1$, $\mathrm{reffwhm} = 2, 2, 0, 0$, $\mathrm{lambda} = 100, 50, 20, 5$, $\mathrm{estint} = 0, 0, 0, 0$, $\mathrm{warpres} = 2, 2, 2$ with spline interpolation (parameters not mentioned here were the defaults as set in FSL 5.0.9).

To compare in vivo with postmortem MRI and histology data, we projected the averaged SI-T1w, T1w, T2*w and PDw images to the MNI reference space (ICBM 152 2009b non-linear symmetric, 500 µm iso. ; *Fonov et al., 2009*; *Fonov et al., 2011*; http://www.bic.mni.mcgill.ca/ServicesAtlases/ICBM152NLin2009). The ICBM 152 reference includes T1w, T2w and PDw data and projecting in vivo and postmortem MRI as well as histology data to this space allowed us also to evaluate the contrast that these commonly used template images have in subcortical auditory areas. To register our in vivo MRI data set to MNI, we used FSL-FNIRT but this time driven by the T1w images (available both in our data set and in the MNI ICBM 152 2009b data).

The postmortem diffusion b0 image was transformed to the post mortem anatomical image space with an affine transformation in ANTs. Anatomical-space images (including the manually segmented atlas) could then be transformed into diffusion space using the 'antsApplyTransforms' command, with the affine transform matrix, a super-sampled diffusion image (from 200 µm to 50 µm to match the anatomical image resolution) as the reference image, and denoting the warp as an inverse transform.

In vivo and postmortem images were registered non-linearly using ANTs. The in vivo SI-T1w image was warped to the postmortem diffusion b0 image following a rigid, then affine, then non-linear SyN algorithm. This produced an in vivo brainstem image in postmortem diffusion space.

The ANTs non-linear registration also created warp and inverse warp transforms that could then be used to transform atlases from one space to another. To preserve the higher resolution of the post mortem MRI when inverse warping postmortem images to in vivo space, we supersampled the in vivo SI-T1w image to 200 µm (matching the post mortem diffusion image) or 50 µm (matching the postmortem anatomical image).

Finally, to transform the postmortem anatomical image (50 µm) to MNI space, we applied the inverse transform from postmortem anatomical to diffusion space (resampled to 50 µm), then the inverse transform from diffusion space to in vivo space (similarly upsampled to 50 µm), and finally from in vivo space to MNI space using the FSL-FNIRT inverse transform (described above).

## BigBrain histology segmentation

In what follows we describe the main anatomical observations related to the auditory structures as segmented in the 100 µm histological data. Images were segmented independently by two raters

(KRS, OFG). Overlap between the two raters was high (see Table 2 [top row - Big Brain across seg-menters] in *Figure 2*); in the figures we show the regions that were consistently segmented by both raters.

## Vestibulocochlear nerve

The vestibulocochlear nerve (the eighth cranial nerve, or CNVIII) enters the brainstem where the medulla and the pons meet (the pontomedullary junction). The cochlear component of the vestibulo-cochlear nerve is composed of spiral ganglion neurons, whose cell bodies are within the cochlea and which carry frequency-specific information to the brainstem.

In the BigBrain histology, CNVIII extends primarily laterally (but also anteriorly and inferiorly) from the pontomedullary junction, bound posteriorly by the cerebellum. Parts of the nerve root are still visible in the images although being cut. It is therefore not labeled in our histological atlas (but see the post mortem MRI atlas below).

## Cochlear nucleus

Once reaching the brainstem, the auditory nerves split into two main routes-one to the anterior ven-tral cochlear nucleus (AVCN), and one to the posterior ventral cochlear nucleus (PVCN) and then on to the dorsal cochlear nucleus (DCN) (*Webster, 1992*). Within each subnucleus, the neurons main-tain the tonotopic frequency representation they receive from the cochlea via the cochlear nerve (*De No, 1933a*; *De No, 1933b*; *Rose et al., 1960*; *Sando, 1965*; *Evans, 1975*; *Ryugo and May, 1993*; *Ryugo and Parks, 2003*) (see bottom panels of the two left most columns in *Figure 2*).

In the BigBrain data, the AVCN is situated anterior and medial to the root of CNVIII, while the PVCN continues from the root of CNVIII and extends posteriorly toward the DCN. The DCN is clearly visible as a dark band wrapping around the cerebellar peduncle posteriorly, becoming exposed on the dorsal surface of the pons.

## Superior olivary complex

The next structure along the auditory pathway is the superior olivary complex (SOC), which in humans is located in the inferior pons. The SOC receives the majority of its ascending inputs from the contralateral cochlear nucleus, although it also receives ipsilateral inputs as well. The contralat-eral dominance is maintained throughout the remaining ascending pathway. The SOC is comprised of the lateral superior olive (LSO), medial superior olive (MSO), and the medial nucleus of the trape-zoid body (MNTB). The size of each of these nuclei varies between species, and it is debated whether the trapezoid body exists in the human SOC (*Moore, 1987*; *Strominger and Hurwitz, 1976*; but see *Kulesza and Grothe, 2015* review of recent findings affirming the existence of the human MNTB).

Although the individual substructures within the SOC have unique anatomy that can be identified from histology (*Moore, 1987*; *Kulesza, 2007*), here we outline the structure of the SOC as a whole in order to include all identifiable substructures (namely the MSO and LSO - see second panel from the bottom of the two left most columns in *Figure 1*). The MSO is the largest SOC nucleus in humans, unlike in other animals. The MSO receives inputs from both the left and right AVCN and sends outputs to the ipsilateral lateral lemniscus. The LSO receives inputs from the ipsilateral AVCN and from the ipsilateral MNTB. Outputs are sent to both ipsilateral and contralateral lateral lemnisci. The MNTB receives inputs from the contralateral AVCN, and its axons terminate in the ipsilateral LSO.

The MSO and LSO are visible in the BigBrain images, despite their small size. The MSO is a thin pencil-like collection of nuclei whose caudalmost point begins around the same axial plane as the rostralmost extent of the AVCN, about 4 mm medial (and slightly anterior) to the AVCN. It then extends about 1 cm rostrally (angled slightly laterally), where it eventually meets the lateral lemniscal tract. The LSO neighbors the MSO near its caudalmost portion, forming a 'V' shape when viewed axially. In our histological atlas, these two structures are combined into a single SOC segmentation. Cells of the MNTB are not clear to us in this sample, so we do not segment it in our atlas.

## Inferior colliculus

The inferior colliculus (IC) is a large, spherical structure in the dorsal midbrain and receives ascending inputs from the auditory brainstem via the lateral lemniscus (see second panel from the top of the two left most columns in *Figure 1*). The central nucleus of the inferior colliculus receives most of these connections, with external nuclei primarily receiving descending connections (*Webster, 1992*). The inferior colliculus sends axons to the medial geniculate body of the thalamus via the brachium of the inferior colliculus.

In the BigBrain data, the inferior colliculus is clearly identifiable as the lower two of the four bumps along the dorsal portion of the midbrain (or tectum). The darkest staining within these structures corresponds to the central nucleus of the inferior colliculus. An intensity gradient outside of the central nucleus likely corresponds to the external and dorsal nuclei, which were included in our segmentation of the IC. Bounding the IC superiorly is the superior colliculus; medially, the commissure of the IC connecting the two inferior colliculi, as well as the aqueduct and periaqueductal grey; and anteriorly, other midbrain nuclei such as the cuneiform nucleus (lateral and inferior to the IC are the borders of the midbrain).

## Medial geniculate of the thalamus

The medial geniculate body (MGB) of the thalamus is the final subcortical auditory structure that sends auditory signals to the auditory cortex via the acoustic radiations (*Winer, 1984*; see top panel of the two left most columns in *Figure 1*). The MGB contains two or three major subdivisions: the ventral MGB receives the majority of IC inputs, while the dorsal and medial subdivisions (at times grouped together, at times separately) receive more varied inputs from auditory and non-auditory subcortical structures.

In the BigBrain sample, the MGB is visible as a dark patch medial to the lateral geniculate nucleus (which can be easily identified by its striations) in a coronal view. Axially, the MGB takes an ovoid shape with a clear dorsolateral boundary next to the brachium of the superior colliculus, which appears light due to lack of cell nuclei being stained. Ventromedially, the MGB is bordered by a light band corresponding to the medial lemniscus. Rostrally, we marked the edge of the MGB where cell staining decreases, at the border with the pulvinar nucleus and ventral posterolateral nucleus of the thalamus.

## Postmortem MRI segmentation

In what follows we describe the anatomical contrast that can be leveraged from these post mortem MRI data in order to identify structures in the auditory brainstem. We then used these segmentations to create an MRI-based atlas of the subcortical auditory system, separate from the BigBrain histology-based atlas.

## Vestibulocochlear nerve

The CNVIII is visible in the post mortem MRI near the pontomedullary junction, extending laterally and anteriorly from the brainstem (see the lower panels in *Figure 2*).

## Cochlear nucleus

The cochlear nuclei are challenging to identify in the postmortem MRI data, although the presence of the CNVIII root provides a landmark for localizing the other structures. Due to low signal contrast around the ventral cochlear nucleus area in the T2*-weighted GRE MRI, we segmented the VCN according to the literature: bound by the cochlear nerve root and wall of the pons laterally, and by cerebellar white matter tracks medially. We were able to segment the dorsal cochlear nucleus based on the T2*-weighted image, where it appears brighter and can be identified as running posteriorly from the VCN and dorsally along the surface of the pons, distal to the inferior cerebellar peduncle.

## Superior olivary complex

As with the cochlear nuclei, the SOC are more difficult to identify in the post mortem MRI than in the histology, likely since the individual subnuclei like the MSO and LSO approach the size of a voxel in at least one direction and are therefore prone to partial voluming effects. However, the pencil-like MSO can still be identified in the coronal plane as a dark, elongated structure in the T2*-weighted

image, starting around the level of the ventral cochlear nucleus. In the axial plane, the SOC (but not its individual subnuclei) can be seen as a dark spot in the T2*-weighted image between the facial nucleus and the trapezoid body (see the second row from the bottom in *Figure 2*).

## Inferior colliculus

As in the BigBrain data, the inferior colliculus is relatively easy to identify based on its gross anatomical structure on the dorsal aspect of the midbrain. Additionally, the MR contrast provides relatively clear boundaries between the colliculi and surrounding structures. Indeed, it may even be possible to segment the inferior colliculus into its subnuclei-the central, external, and dorsal nuclei-based on T2*-weighted MR signal intensities (see the second row from the top in *Figure 2*). The external nucleus of the IC appears dark in the T2*-weighted image, on the lateral aspect of the IC. Medial to the external nucleus is the central nucleus, which has higher T2*-weighted intensity (appears brighter) in our MR images, and has clear boundaries on its ventral, medial, and dorsolateral sides. The dorsal nucleus is along the dorsal aspect of the IC and is the brightest subcomponent within the IC in terms of T2*-weighted MR signal.

## Medial geniculate

Although the borders of the MGB are less clear in the post mortem MRI than in the BigBrain images, the structure itself is again relatively easy to identify by its gross anatomical location as well as MR signal intensity. In the coronal plane, the medial geniculate is medial to the lateral geniculate at the junction of the midbrain and thalamus. Axially, the medial geniculate has circular or ovoid shape, again medial to the lateral geniculate. In the axial plane, the medial geniculate is largely bordered dorsolaterally by the brachium of the superior colliculus, which appears as a thick, dark band of fibers in the T2*-weighted image. Medially, the medial geniculate is bound by the brachium of the inferior colliculus (also appearing as a dark fiber band), at least through the caudal half of the structure. We have included the portions of this fiber bundle in the segmentation of the medial geniculate, as the auditory fibers connecting the IC and the MGB are quite relevant to MRI connectivity investigations (including our own; post mortem tractography results below).

As with the inferior colliculus, it may be possible to identify separate divisions within the medial geniculate. Within the overall structure, there are two identifiable substructures based on T2*-weighted MR image intensity. Dorsomedially (and somewhat caudally), about half of the medial geniculate has high T2*-weighted contrast and appears bright; the ventrolateral (and slightly rostral) half appears darker in the T2*-weighted image. These segmentations largely (but not perfectly) align with the ventral and dorsal/medial nuclei of the medial geniculate in the Allen Human Brain Atlas (*Hawrylycz et al., 2012*), as well as with those of *Paxinos (2019)*. However, they vary somewhat from the the axial slice segmentation from *Merker (1983)* shown in *Amunts et al. (2012)*, which show a largely horizontal delineation between the substructures.

## Functional MRI analysis

In both functional experiments, data were preprocessed using BrainvoyagerQX version 2.8.4 (*Goebel, 2012*). Slice-scan-time correction, motion correction, temporal high-pass filtering (GLM-Fourier, six sines/cosines) and temporal smoothing (Gaussian, width of kernel 5.2 s). The defaults in BrainvoyagerQX v2.8.4 were used for these steps aside from the explicitly stated values. The functional images were then distortion corrected using the opposite phase encoding direction images using FSL-TOPUP (*Andersson et al., 2003*). Conversion between Brainvoyager file types to NIfTI which was required to perform distortion correction was done using Neuroelf version 1.1 (release candidate 2) http://neuroelf.net/ in Matlab version 2016a. For alignment across experiments (i.e. to co-register the data of experiment two to the ones collected in experiment 1) we used FSL-FLIRT. In this procedure the alignment between the functional data of the two experiments was tailored to a mask that included the brainstem, thalamus and auditory cortex.

After pre-processing, functional images were then transformed to Talairach space using Brainvoyager at a resolution of 0.5 mm isotropic. We have previously used this procedure in order to reveal tonotopic maps in both the inferior colliculus and medial geniculate nucleus (*De Martino et al., 2013*; *Moerel et al., 2015*) and have shown that the upsampling has no consequence on the spatial distribution of the responses. Upsampling can also reduce effects of interpolation that is common

during resampling in many image processing steps. After upsampling, mild spatial smoothing (Gaussian, FWHM 1.5 mm) was also applied. *Figure 4—figure supplement 5* shows the effect that spatial smoothing has on the activation maps obtained from two participants data in experiment 1.

GLM-denoise (*Kay et al., 2013*) was used to estimate noise regressors. In brief, for each cross validation a noise pool of non responsive voxels (i.e. voxels with a response to sound representation determined by an F-statistic below a given threshold) was determined on the training data set (16 runs across the two sessions of experiment 1 and 12 runs across the two sessions of experiment 2) and used to obtain noise regressors defined as the principal components of the noise pool time course matrix that added to a GLM analysis (*Friston et al., 1994*) of the training data would result in an increased activation. The number of noise regressors was optimized using cross validation within the training set. The selected noise regressor spatial maps were projected on the test data to obtain the regressors for the test data.

Similarly, the hemodynamic response function (HRF) best characterizing the response of each voxel in the brainstem was obtained using a deconvolution GLM (with nine stick predictors) on the training data. Note that this procedure, while possibly overfitting information in the training data, produces noise regressors and an HRF for each test run (e.g. the noise regressors for runs 4, 6 and 9 of session one in experiment 1 comes from an analysis performed on all other runs in the same session) that are not overfitted.

The resulting HRF and noise regressors were used in a GLM analysis of the test runs. We combined all test runs (for each individual voxel) using a fixed effect analysis.

Statistical maps of responses to sounds vs silence were corrected for multiple comparisons at the individual level using False Discovery Rate (FDR; q-FDR = 0.05). An additional threshold on the uncorrected p-value of each voxel (i.e., $p<0.001$) was applied to further reduce the number of false positive activation that can be expected when applying FDR. Unless otherwise stated, single subject statistical maps are displayed by color coding voxels that surpass these statistical thresholds. Unthresholded statistical maps are visualized in Figure 10 and are available at the online repository of the data (https://osf.io/hxekn/) for inspection.

The functional activation maps of the six participants that took part in both experiments have been analyzed to demonstrate within participant reproducibility of effects. Since the stimuli were different and the number of runs were different, this second experiment shows a generalization of the first experiment, thereby additionally validating the detection of these structures. *Figure 4—figure supplement 3* shows the statistically thresholded activation maps for each of this six participants for the two experiments in three anatomical cuts (two transversal for CN/SOC and IC and one coronal for the MGB). The percentage of statistically significant voxels in experiment one that are statistically significant in experiment two is reported together with the distance between the centroids of activations between the two experiments in *Figure 4—figure supplement 4* (for each individual and in average across individuals). The unthresholded maps of both experiments (for each of the six participants) are also visualized in Figure 11 and are available at the online repository (https://osf.io/hxekn/) for inspection.

To produce group level results, the single subject statistical maps were warped to the reference brainstem (subject 1) by applying the warping field obtained on the anatomical data. After projection to the common space, single subject statistical maps were binarized and converted to a probabilistic map by: 1) applying of a cluster size threshold of 3.37 mm$^3$ (27 voxels in the 0.5 mm isotropic anatomical space 2.5 voxels in the original functional resolution) and 2) summing maps across subjects at each single voxel (i.e. a value of 10 indicates that all 10 subjects exhibited a statistically significant response to sound presentation corrected for multiple comparisons and belonging to a cluster of at least 27 voxels in the anatomical space). The additional clustering allowed us to further control for possible false positives by imposing a neuroanatomically plausible hypothesis (i.e. none of our region of interest is smaller than 3.37 mm$^3$ in volume). The same procedure was also repeated by leaving one subject out (i.e. we generated probabilistic maps from 9 out of the 10 subjects each time leave one subject out). The leave-one-out probabilistic maps were then back-projected to the anatomical space of the left out subject (i.e. the probabilistic map obtained from subjects 1 to 9 was back-projected to the anatomical space of subject 10). Unless otherwise stated, probabilistic maps are displayed with minimum threshold of at least three out of 10 (or nine for the leave one out maps) subjects exhibiting significant responses at each voxel. Unthresholded probabilistic maps are available for inspection at the online repository.

We evaluated how well cluster localized on the basis of our probabilistic maps generalize to new data. *Figure 4* displays the statistically thresholded activation maps for each of the ten participants in experiment one in three anatomical cuts (two transversal for CN/SOC and IC and one coronal for the MGB) together with the probabilistic map obtained from the other nine participants (thresholded by displaying voxels that are functionally significant in at least three out of nine participants). In *Figure 4—figure supplement 1*, we report the percentage of voxels in the leave one out probabilistic maps that are statistically significant in the left out subject. The overlap is reported toegther with the distance between the centroids of activations in the leave one out probabilistic maps and the left out subject. The effect of the threshold on the probabilistic maps is analyzed in *Figure 4— figure supplement 2*. The unthresholded maps (leave one subject out and single subject) are also visualized in Figure 10 and available at the online repository (https://osf.io/hxekn/) for inspection.

To compare the functional activation maps with histology data and post mortem MRI data, the probabilistic maps were projected to the MNI space using the warping field obtained from the anatomical dataset.

## BigBrain data

Histology data were obtained by downloading the 100 µm version of the BigBrain (*Amunts et al., 2013*) 3-D Volume Data Release 2015 (from https://bigbrain.loris.ca). We downloaded both the original images and the dataset already aligned to MNI ICBM 152. The nuclei along the auditory pathway (cochlear nucleus, superior olive, inferior colliculus and medial geniculate nucleus) were manually segmented in the histology space image using ITK-SNAP (*Yushkevich et al., 2006*) largely following the definitions in *Moore (1987)* when possible.

### Correction of the alignment of the inferior colliculi to MNI

Upon visual inspection of the BigBrain image in the MNI ICM 152 space, we detected a major registration error around the inferior colliculi (see *Figure 7* - second panel from the left). The registration quality to MNI ICMBM 152 space in the rest of the brainstem was deemed satisfactory, but the the region of the inferior colliculus required correction in order to perform a valid comparison with the MRI data (in vivo and post mortem). Interestingly, the region of the colliculi of the BigBrain in the original histology space appeared to be closer in location to the position of the inferior colliculus in the MNI dataset (compare panels 1 and 3 in *Figure 7*) indicating that the highlighted misalignment in the original BigBrain MNI dataset originated during the registration procedure.

To perform a new registration to MNI of the brainstem and thalamus of the BigBrain data that observed the already correctly registered boundaries (e.g. the Pons) but corrected the region around the inferior colliculus bilaterally, we followed N steps. First, we defined a region of interest around the inferior colliculus using common anatomical landmarks that were visible in the BigBrain MNI and MNI (2009b) T1, PD, T2 images and where aligned satisfactorily. Second, this region was cut out from the BigBrain MNI and replaced by the same region (i.e. defined by the same anatomical landmarks) in the BigBrain histology space data (before projection to MNI). The convex hulls of the region of interest in the BigBrain histology and in the MNI space were matched using 3-D optimal transport as implemented in Geogram version 1.6.7 (*Lévy, 2015*; *Lévy and Schwindt, 2018*). Third, the convex hull matched region of the the BigBrain histology space was used to replace the incorrect region which was cut out at step 2. As a result of these three steps we obtained a version of the BigBrain in MNI (BigBrain MNI - implanted) that had the inferior colliculus in the right position but where the transitions between outside to inside of the region of interest that was corrected were visible and not respecting of the topology. To correct for these residual errors, we performed a new FSL-FNIRT alignment between the original BigBrain in histology space and the BigBrain MNI - implanted image. The resulting image (BigBrain MNI - corrected) preserved the actual topology inside the brainstem and at the same time resulted in a correct alignment of the regions around the inferior colliculus bilaterally (see *Figure 7* - right panel).

## Postmortem MRI vasculature analysis

Gradient echo (GRE) MRI is sensitive to vasculature within the imaged tissue. To highlight vasculature in the post mortem brainstem specimen, we computed the minimum intensity projection in coronal sagittal and axial direction from the 50 µm isotropic voxel GRE MRI data over slabs of 1.1 mm

in thickness using Nibabel (*Brett et al., 2017*) and Numpy (*van der Walt et al., 2011*). This image can be seen in *Figure 5* right column.

## Diffusion MRI analysis

### Postmortem diffusion

Before analysis, postmortem diffusion volumes were each registered to the first b0 volume using an affine transformation in ANTs version 2.1.0 (*Avants et al., 2011*). To estimate white matter fiber orientations, we used the constrained spherical deconvolution (CSD) model as implemented in DIPY 0.14 (*Gorgolewski et al., 2011*; *Garyfallidis et al., 2014*; *Tournier et al., 2007*) as a Nipype pipeline (*Gorgolewski et al., 2011*). CSD posits that the observed diffusion signal is a convolution of the true fiber orientation distribution (FOD) with a response function. DIPY's 'auto-response' function estimates the fiber response function from a sphere of 10 voxels in the center of the sample above a given fractional anisotropy (FA) threshold (0.5 in our study). We then estimated FOD peaks in each voxel using DIPY's 'peaks-from-model' method with a 10° minimum separation angle and a maximum of 5 peaks per voxel.

White matter fiber streamlines were estimated deterministically with DIPY's EudX method (*Mori et al., 1999*; *Garyfallidis, 2013*) with 1,000,000 seeds per voxel, a 75° streamline angle threshold, and an FA termination threshold of 0.001 (since data outside the specimen sample were already masked to 0).

To define regions of interest (ROIs) for the fiber display, the auditory structures manually delineated in the post mortem T2*-weighted MR images were transformed to diffusion space using ANTs, and global streamlines were filtered by considering only the voxels in each one of the ROIs as a seed and further constrained by using all auditory ROIs as tractography waypoints. This resulted in a high-resolution, high-quality auditory-specific subcortical tractogram, which were then visualized in TrackVis 0.6.1 (*Wang et al., 2007*).

### In vivo diffusion

7T in vivo dMRI data was corrected for distortions with the HCP pipeline (*Glasser et al., 2016*; *Sotiropoulos et al., 2013*). Specifically, geometric and eddy-current distortions, as well as head motion, were corrected by modeling and combining data acquired with opposite phase encoding directions (*Andersson et al., 2003*; *Andersson and Sotiropoulos, 2015*; *De No, 2016*). The data were then masked to include just the brainstem and thalamus, matching the post mortem specimen.

Similar to the post mortem analysis, we estimated diffusion FODs with a CSD model implemented in DIPY with response function FA threshold of 0.5. Peaks were extracted with a minimum separation angle of 25°. White matter connectivity was estimated with deterministic tractography throughout the brainstem and thalamus, again using DIPY's EudX algorithm (*Mori et al., 1999*; *Garyfallidis, 2013*) with 1,000,000 seeds per voxel, a 45° streamline angle threshold, and an FA termination threshold of 0.023.

For the tractography in the in vivo data we used subcortical auditory ROIs as defined by the analysis of the functional data (i.e. regions that exhibited significant [corrected for multiple comparisons] response to sound presentation in at least three out of 10 subjects). The functional ROIs were transformed to individual diffusion space and used as tractography seeds, with all other auditory ROIs as waypoints, producing a subcortical auditory tractogram for each in vivo subject.

## Data and code availability

Unprocessed in vivo data are available at https://openneuro.org/datasets/ds001942/versions/1.2.0 (DOI: 10.18112/openneuro.ds001942.v1.2). Atlas segmentations and tractography streamlines are available through the Open Science Framework (https://osf.io/hxekn/). Processing and analysis resources, including links to all data and software used in this paper, are available at https://github.com/sitek/subcortical-auditory-atlas (*Sitek and Gulban, 2019*; copy archived at https://github.com/elifesciences-publications/subcortical-auditory-atlas). See *Figure 8* for an overview of currently available data and code (full resolution version available at our code repository).

## Animated 3D volume renderings

Video animations in *Figure 9*, *Figure 10* and *Figure 11* were created using pyqtgraph (v0.10.0, http://www.pyqtgraph.org/) volume rendering. The t-value maps were clipped to 0–20 range and scaled to 0–255 range. These t-values are 3D volume rendered by assigning the corresponding gray value to each voxel as well as the alpha channel (transparency). Which means that lower values are closer to black and translucent. Animation frames were generated by rotating camera one degree at a time for 360 degrees. Additive rendering was used for 2D projections to provide depth vision (i.e. for preventing voxels closest to the camera from seeing values inside the clusters.).

## Additional information

### Funding

| Funder | Grant reference number | Author |
|---|---|---|
| Nederlandse Organisatie voor Wetenschappelijk Onderzoek | 864-13-012 | Omer Faruk Gulban Federico de Martino |
| National Institutes of Health | 5R01EB020740 | Satrajit S Ghosh |
| National Institutes of Health | P41EB019936 | Satrajit S Ghosh |
| National Institutes of Health | 5F31DC015695 | Kevin R Sitek |
| Eaton Peabody Laboratory at Mass Eye and Ear | Amelia Peabody Scholarship | Kevin R Sitek |
| Harvard Brain Science Initiative | Travel Grant | Kevin R Sitek |
| National Institutes of Health | P41EB015897 | G Allan Johnson |
| National Institutes of Health | 1S10OD010683-01 | G Allan Johnson |

The funders had no role in study design, data collection and interpretation, or the decision to submit the work for publication.

### Author contributions

Kevin R Sitek, Conceptualization, Resources, Data curation, Software, Formal analysis, Validation, Investigation, Visualization, Writing—original draft, Writing—review and editing; Omer Faruk Gulban, Conceptualization, Resources, Data curation, Software, Formal analysis, Validation, Investigation, Visualization, Methodology, Writing—original draft, Writing—review and editing; Evan Calabrese, Resources, Investigation, Methodology, Writing—review and editing; G Allan Johnson, Resources, Data curation, Funding acquisition, Investigation, Methodology, Writing—review and editing; Agustin Lage-Castellanos, Software, Formal analysis, Validation, Visualization; Michelle Moerel, Data curation, Methodology, Resources; Satrajit S Ghosh, Federico De Martino, Conceptualization, Resources, Supervision, Funding acquisition, Methodology, Writing—review and editing

### Author ORCIDs

Kevin R Sitek (iD) https://orcid.org/0000-0002-2172-5786
Omer Faruk Gulban (iD) https://orcid.org/0000-0001-7761-3727
G Allan Johnson (iD) https://orcid.org/0000-0002-7606-5447
Satrajit S Ghosh (iD) https://orcid.org/0000-0002-5312-6729

### Ethics

Human subjects: The experimental procedures were approved by the ethics committee of the Faculty for Psychology and Neuroscience at Maastricht University (reference number: ERCPN-167_09_05_2016), and were performed in accordance with the approved guidelines and the Declaration of Helsinki. Written informed consent was obtained for every participant before conducting the experiments. All participants reported to have normal hearing, had no history of hearing disorder/impairments or neurological disease.

**Decision letter and Author response**
Decision letter https://doi.org/10.7554/eLife.48932.055
Author response https://doi.org/10.7554/eLife.48932.056

## Additional files

### Supplementary files
• Transparent reporting form
DOI: https://doi.org/10.7554/eLife.48932.044

### Data availability

In vivo data are available on OpenNeuro: https://openneuro.org/datasets/ds001942. Derivatives (including histology-based, post mortem MRI-based, and fMRI-based atlases) are available on the Open Science Framework: https://osf.io/c4m82/. Analysis code, flowcharts, and other auxiliary files are available on Github: https://github.com/sitek/subcortical-auditory-atlas (copy archived at https://github.com/elifesciences-publications/subcortical-auditory-atlas).

The following datasets were generated:

| Author(s) | Year | Dataset title | Dataset URL | Database and Identifier |
|---|---|---|---|---|
| Omer Faruk Gulban, Kevin R Sitek, Satrajit S Ghosh, Michelle Moerel, Federico De Martino | 2019 | Auditory localization with 7T fMRI | https://openneuro.org/datasets/ds001942/versions/1.2.0 | OpenNeuro, 10.18112/openneuro.ds001942.v1.2.0 |
| Sitek KR, Gulban OF, Calabrese E, Johnson GA, Lage-Castellanos A, Moerel M, Ghosh SS, De Martino F | 2019 | Mapping the human subcortical auditory system | https://doi.org/10.17605/OSF.IO/HXEKN | Open Science Framework, 10.17605/OSF.IO/HXEKN |

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

# Appendix 1

DOI: https://doi.org/10.7554/eLife.48932.045

## Glossary

### Anatomical abbreviations

| | |
|---|---|
| **AVCN** | **Anteroventral cochlear nucleus.** |
| CN | Cochlear nucleus. |
| CNVIII | 8th nerve, vestibulocochlear nerve. |
| DCN | Dorsal cochclear nucleus. |
| IC | Inferior colliculus. |
| LGN | Lateral geniculate nucleus. |
| LSO | Lateral superior olive. |
| MGB/MGN | Medial geniculate body/nucleus. |
| MNTB | Medial nucleus of the trapezoid body. |
| MSO | Medial superior olive. |
| PVCN | Posteroventral cochlear nucleus. |
| SOC | Superior olivary complex. |

### MRI acquisition abbreviations

| | |
|---|---|
| **7T** | **7 Tesla.** |
| dMRI | diffusion magnetic resonance imaging. |
| FOV | Field of view. |
| fMRI | functional magnetic resonance imaging. |
| GRAPPA | Generalized auto-calibrating partially parallel acquisitions. |
| MB | Multi-band. |
| MPRAGE | Magnetization prepared rapid acquisition gradient echo. |
| MRI | Magnetic resonance imaging. |
| PDw | Proton density weighted. |
| SI-T1w | Short inversion time T1-weighted. |
| T1w | T1-weighted. |
| T2*w | T2*-weighted. |
| TE | Echo time. |
| TR | Repetition time. |

### Data analysis abbreviations

| | |
|---|---|
| **CSD** | **Constrained spherical deconvolution.** |
| FA | Fractional anisotropy. |
| FDR | False discovery rate. |

*continued*

| CSD | Constrained spherical deconvolution. |
|---|---|
| FOD | Fiber orientation distribution. |
| GLM | General linear model. |
| HCP | Human connectome project. |
| HRF | Hemodynamic response function. |
| ICBM | Internation Consortium for Brain Mapping. |
| M0 | T2 signal with no diffusion weighting. |
| MD | Mean diffusivity. |
| MNI | Montreal Neurological Institude. |
| MSMT | Multi-shell multi-tissue |
| ODFs | Orientation distribution functions. |
| ROI | Region of interest. |

