## [Decision Letter]

[Editors’ note: a previous version of this study was rejected after peer review, but the authors submitted for reconsideration. The first decision letter after peer review is shown below.]

Thank you for submitting your work entitled "Mapping the human subcortical auditory system using histology, post mortem MRI and in vivo MRI at 7T" for consideration by *eLife*. Your article has been reviewed by four peer reviewers, one of whom is a member of our Board of Reviewing Editors, and the evaluation has been overseen by a Senior Editor. The following individual involved in review of your submission has agreed to reveal their identity: Marta Correia (Reviewer #4).

Our decision has been reached after consultation between the reviewers. Based on these discussions and the individual reviews below, we regret to inform you that your work will not be considered further for publication in *eLife*. There was widespread appreciation of the amount of care and work that went into the analyses, and many of the results are impressive, particularly the ex vivo work. At the same time the in vivo part seemed less clear and potentially underpowered, and there was general concern about the correspondence between function and anatomy, especially for the lower brainstem structures.

If you are able to substantially add to the in vivo story and (as you've indicated) provide all of the maps for the broader community, we would consider a new submission under "tools and resources". However, this would need to be a substantially improved paper and would likely go to the same set of reviewers.

Reviewer #1:

This is a very nicely done atlas of subcortical auditory structures that includes histological identification and connectivity from MRI (post mortem and in vivo). Although the results are large confirmatory, this is a thoroughly impressive technical achievement and the release and publication of these atlases is likely to be impactful.

That being said, I understand the reasons for not making the segmentations and streamlines available yet, but the work is difficult to fully judge without seeing these (for example, to determine how accessible the formats are for other researchers). I think the data and code will need to be made available to the reviewers in some fashion before the work could be accepted.

Reviewer #2:

This is a helpful comparison of histological, postmortem MR, and in-vivo structural and functional approaches to localizing auditory brainstem and midbrain structures. The authors make use of previously acquired data and public atlases to align estimates of cochlear nuclei, olivary nuclei, inferior colliculi and medial geniculate nuclei across these different datasets, along with estimating the tract connections between them using DTI acquired ex- and in-vivo.

Clearly a lot of work has gone into the various components of this project and care has been taken with the analyses. What wasn't clear to me was what – if anything – was the 'message' of the paper, except maybe one that was not intended, e.g., that even when state-of-the-art techniques are used on high quality imaging data, it is extremely difficult to identify smaller auditory nuclei accurately in-vivo. Indeed, even the connectivity of the IC is not straightforward to map out despite much higher quality data than is usually available along with generally careful processing.

My major concerns were as follows (note that the authors do mention the majority of these as issues in the Discussion):

1) The fMRI-defined nuclei are clearly inaccurate in that they are up to an order of magnitude larger than the actual anatomical structures, and do not conform to the structures' shape. This must stem in part from the surprising decision to use a 3.3mm FWHM smoothing kernel on high-resolution data. It was also surprising that no unmasked group z maps were presented; I can understand the idea behind the conjunction maps (showing x of 10 subject with FDR-corrected activation there), but the pretty minimal overlap in the SOC (3 of 10 subjects) doesn't inspire too much confidence. I also wondered where the cluster threshold of 27 voxels came from (subsection “Functional MRI analysis”, last paragraph), and what the rest of the slices look like through the brainstem (e.g., how many splotches are there).

2) The similarity between the estimates of CN and SOC across the BigBrain, ex-vivo, and in-vivo segmentations is near or at zero (as shown by the dice coefficients). Since defining these smaller structures is really the main potential contribution of the paper (given IC and MGB have been previously identified by a number of groups), this is concerning.

3) The tractography results also essentially show that the known connections between any of the nuclei cannot be reliably reproduced, especially in-vivo. For instance, in the demonstration subject in Figure 6 as well as the matrix showing total streamlines, the IC is essentially being bypassed, despite the fact that it is (to my knowledge) the major source of input to the MGB. In the Figure 3—figure supplement 1 of the postmortem data, there are also a lot of streamlines that must be incorrect (most prominently, the large set of streamlines going inferiorly, but also the huge projections from around the SOC and CN seeds that seem to bypass the IC and go directly into the MGB). The large projections going inferiorly from the IOC seeds (Figure 4) are also rather mysterious. Particularly given the statements regarding the 'vastly improved estimation of white matter connections'.

In terms of overall contribution, having labelled nuclei in two publicly available postmortem datasets is helpful, and the fix to the incorrect MNI warp of bigbrain is also good to have. I'm less convinced about the contribution of the fMRI data (compared to the previously published studies that they come from), due to the way that they were processed and analyzed. If anything, the DTI tractography results seem to confirm the problems that Thomas et al., 2014, showed with even better diffusion data.

Reviewer #3:

This manuscript examines the structure of human auditory subcortical structures by recording in-vivo 7T functional magnetic resonance imaging (fMRI), structural MRI, and diffusion MRI (N=10). Ex-vivo 7T structural and diffusion MRI were also recorded in a single post-mortem subject. A subcortical atlas was created from the post-mortem structural MRI and publicly available histology data, and used to validate the functionally identified subcortical regions.

All four subcortical auditory structures (MGN, IC, SOC, CN) were successfully identified using ex-vivo structural MRI, while in-vivo structural MRI could only be used to identify the MGN and IC. Using functional data, the MGN and IC could be identified in majority of the participants, while the SOC and CN could only be identified in a small subset participants (n = 2-3). Ex-vivo tractography showed significant within-structure streamlines while the number of between-structure streamlines was quite modest. On the other hand, in-vivo diffusion imaging revealed significant streamlines between and within structures.

In general, the paper makes a useful contribution to the literature, and the figures are lovely. The Introduction, however, could be reorganized to more clearly indicate the gap in knowledge and the research question that this work was designed to address, since the general location of the main subcortical auditory processing nuclei is already well known in humans. One strength of the paper is that these findings replicate the authors' previous work (cited Discussion, ninth paragraph)-lending further credence to the idea that subcortical structures can be reliably identified from functional data. However, one limitation is that functional localization was not very strong in lower brainstem nuclei (i.e., SOC and CN) for majority of the participants (cf. Figure 5 left and middle panel and Figure 5—figure supplement 1). I do wonder if the in-vivo portion of the study is under powered. This issue could be addressed using additional commentary by the authors, although if the N is not supported by a power analysis (or some other logical explanation), the paper could be improved by recording data from a larger sample.

Reviewer #4:

I was asked to provide a review of the diffusion MRI methods used in this paper. The tractography methods used follow the current state-of-the-art and were implemented using the well validated tools within dipy. The authors are also cautious with their interpretation of results, in line with the known limitations of tractography methods.

I don't have any major concerns over the methods used.

---

## [Author Response]

[Editors’ note: what now follows is the decision letter after the authors submitted for further consideration.]

Reviewer #1:This is a very nicely done atlas of subcortical auditory structures that includes histological identification and connectivity from MRI (post mortem and in vivo). Although the results are large confirmatory, this is a thoroughly impressive technical achievement and the release and publication of these atlases is likely to be impactful.That being said, I understand the reasons for not making the segmentations and streamlines available yet, but the work is difficult to fully judge without seeing these (for example, to determine how accessible the formats are for other researchers). I think the data and code will need to be made available to the reviewers in some fashion before the work could be accepted.

We appreciate reviewer #1’s comments and recognize that our code and data must be publicly available in order to fully consider our contributions. We have therefore shared our code on Github and our segmentations and streamlines on the Open Science Framework. In addition, original and processed in vivo data are made available in BIDS format through OpenNeuro and NeuroVault. Here are the links to these addresses for convenience: – in vivo data are available on OpenNeuro:

https://openneuro.org/datasets/ds001942/versions/1.2.0 (DOI: 10.18112/openneuro.ds001942.v1.2.0) - Derivatives are available on the Open Science Framework: https://osf.io/c4m82/ - Analysis code, flowcharts and other auxiliary files are available here: https://github.com/sitek/subcortical-auditory-atlas.

Reviewer #2:This is a helpful comparison of histological, postmortem MR, and in-vivo structural and functional approaches to localizing auditory brainstem and midbrain structures. The authors make use of previously acquired data and public atlases to align estimates of cochlear nuclei, olivary nuclei, inferior colliculi and medial geniculate nuclei across these different datasets, along with estimating the tract connections between them using DTI acquired ex- and in-vivo.Clearly a lot of work has gone into the various components of this project and care has been taken with the analyses. What wasn't clear to me was what – if anything – was the 'message' of the paper, except maybe one that was not intended, e.g., that even when state-of-the-art techniques are used on high quality imaging data, it is extremely difficult to identify smaller auditory nuclei accurately in-vivo.

Reviewer #2 raises a valid critique regarding the outcomes of our manuscript in light of the in vivo results. In the revised version of the manuscript we have expanded our in vivo dataset and analyses to demonstrate the reliability of our functional localization within each individual and the generalization across individuals, as well as improved and clarified our group-level functional atlas. The new analyses for the in vivo functional data are reported in the new Figure 4 together with Figure 4—figure supplements 1-5.

In particular, Figure 4 shows single subject statistical maps together with probabilistic maps obtained using a leave-one-subject-out procedure. The overlap between the probabilistic leave-one-subject-out maps and the single subjects is analyzed in Figure 4—figure supplements 1 and 2. This first set of results shows that: a) a cluster of voxels significantly responding to sounds can be identified in each volunteer; b) these clusters co-localize to regions identified probabilistically in the other nine volunteers; and c) the overall response is higher for the IC and MGB and lower for the CN and SOC.

We then turned our attention to demonstrate the within-subject reliability of these functional definitions. Figure 4—figure supplement 1 shows the statistical maps obtained in a second experiment for six of the ten participants. These results are reported together with the overlap between the two experiments statistical maps in Figure 4—figure supplement 2.

Both within subjects and across subjects, the reliability is higher for the IC and MGB than for the CN and SOC. The between-subject reliability (i.e., the lower overlap between subjects at the level of CN and SOC) is most likely affected by the lower accuracy of the anatomical alignment for these structures caused by the absence of anatomical contrast to be used to drive the alignment procedure. This issue is discussed in the manuscript where we state:

“Another factor that may have impacted the increased volume of the in vivo probabilistic regions can be the residual anatomical misalignment across subjects that also contributes especially to the lower degree of overlap at CN and SOC.”

The within-subject reliability is most likely impacted by the lower statistical power in these regions together with residual alignment errors across experiments, which we also discuss in the revised manuscript where we state:

“The higher signal-to-noise ratio in attainable in regions corresponding to the IC and MGB results in highly reproducible functional responses both within and across participants in these regions. Activation clusters identified at the level of CN and SOC in single individuals do reproduce (albeit to a smaller degree with respect to IC and MGB) both within subjects (i.e., across experiments) and across subjects.”

It has to be noted that Experiment 1 and 2 have been conducted with up to three years of difference in time between them. Additionally, the experiments had different numbers of runs and sounds presented per run (24 vs. 16 runs, 168 vs. 96 sounds). The overall reproducibility of the results demonstrates that the functional definition of the auditory nuclei is reliable both within and across subjects, something that we now clarify in the manuscript:

“In each participant we identified voxels significantly responding to sound presentation in regions corresponding to the CN, SOC, IC and MGB. We validated these definition by evaluating both the within-subject reproducibility (i.e., by comparing functional maps across two experiments in six individuals) and the ability of a probabilistic atlas defined on nine out of our ten participants to generalize to the left out volunteer.”

Nevertheless, discrepancies between the in vivo and the post mortem results in terms of the localization and the overlap of the definitions remain. We believe the discussion on the revised manuscript describes what our thoughts are over these discrepancies that can in short can be attributed to: 1) partial volume that still impact the in vivo data; and 2) alignment between post mortem and in vivo rendered cumbersome by the difference (or absence) of anatomical contrast in the relevant ROIs.

Indeed, even the connectivity of the IC is not straightforward to map out despite much higher quality data than is usually available along with generally careful processing.

We agree with the reviewer that tractography remains challenging, particularly between the small, dense, deep brainstem nuclei. We discuss some of the challenges below in more detail. Despite these challenges, we were able to successfully estimate connections between auditory structures (as defined by functional localization in the same group of subjects). A video of the in vivo streamlines in one participant is available as Figure 3—video 1. In fact, the resulting connectivity matrix (Figure 7, top right) closely resembles expected connectivity patterns based on animal research. For instance, there appear to be separate connectivity networks on the left and right above the CN, while there are some commissural connections at the CN, SOC, and IC levels, as expected.

While these connection estimates align with our general thinking regarding subcortical auditory connectivity, there is unfortunately no gold standard for human anatomical connectivity, as tracer methods available in animal research are largely unavailable in humans. However, we are encouraged by the present results, which are consistent across participants (see Figure 6—figure supplement 1), and believe auditory neuroscientists should continue to push for improved DWI acquisition and tractography methods to further hone the methodology to study the subcortical auditory system.

My major concerns were as follows (note that the authors do mention the majority of these as issues in the Discussion):1) The fMRI-defined nuclei are clearly inaccurate in that they are up to an order of magnitude larger than the actual anatomical structures, and do not conform to the structures' shape. This must stem in part from the surprising decision to use a 3.3mm FWHM smoothing kernel on high-resolution data.

We appreciate the reviewer’s careful critique of our methods and have addressed them in our revised manuscript. Regarding spatial smoothing, we feel that the concern of the reviewer stemmed from a misunderstanding caused by the way that the smoothing kernel was reported originally. Namely, we applied 3 voxel FWHM smoothing after transforming the functional data to 0.5 mm iso. resolution in Talairach space. This 3 voxel FWHM spatial smoothing translates to 1.5mm FWHM. We have corrected the reporting in the Materials and methods section where we state:

“…functional images were then transformed to Talairach space using Brainvoyager at a resolution of 0.5 mm isotropic. We have previously used this procedure in order to reveal tonotopic maps in both the inferior colliculus and medial geniculate nucleus (DeMartino, 2013, Moerel, 2015) and have shown that the upsampling has no consequence on the spatial distribution of the responses. After upsampling, mild spatial smoothing (Gaussian, FWHM 1.5mm) was also applied.”

On top of this, we have now added the comparison between unsmoothed and smoothed maps as Figure 4—figure supplement 5. Mildly smoothed functional data helps bringing out the anatomically plausible clusters by improving our SNR, especially in lower brainstem regions of interest.

It was also surprising that no unmasked group z maps were presented; I can understand the idea behind the conjunction maps (showing x of 10 subject with FDR-corrected activation there), but the pretty minimal overlap in the SOC (3 of 10 subjects) doesn't inspire too much confidence.

We interpret this statement to mean that the results for CN/SOC, where there is lower overlap across participants, would be better interpreted looking at unthresholded maps. We agree with the reviewer and think that in order to have a full understanding of our data, a better picture can be gathered by looking at all single individual data as well as the group data (both thresholded and unthresholded).

For this reason, we now report in Figure 4 all single subject statistical maps (thresholded for significance). The unthresholded data are reported in video format for each individual both thresholded and unthresholded (see Figure 10—video 1-10). The single subject data clearly indicate that clusters of significant activity can be identified at the level of CN and SOC in most participants (with the exception of the right SOC in S03, the left CN and SOC in S08 and the left CN in S09). How well regions defined in a group of subjects allow us to localize regions significantly responding to sounds in a new subject was also assessed by conducting a leave-one-out analysis for each participant and calculating the distance and overlap between each participant’s cluster and the cluster comprised of all other participants’ activation. These are shown in Figure 4—figure supplements 1, 2 and in Figure 10—video 1-10. In addition, our unthresholded and thresholded maps are open to inspection here: https://osf.io/c4m82/

See the data used in this figure as 3D volume renders for each 10 subject in Figure 10—videos 1-10 in our updated manuscript, in Figure 10.

The reproducibility of these results was evaluated in test-retest reliability analysis by looking at data collected in a separate experiment in six of the ten participants (reported in Figure 4—figure supplement 3). The distance and overlap between each participant’s clusters of activation across the two experiments are reported in Figure 4—figure supplement 4.

See the data used in this figure as 3D volume renders for each 6 subject in Figure 11—videos 1-7 in our updated manuscript, in Figure 11.

We have also looked at the group average results for further comparison of the activity patterns in experiment 1 and 2. Group averaged 3D volume renders for fMRI activity can be seen in Author response video 1.

**Author response video 1. respvideo1:** Group average fMRI results from two separate experiments.

On top of these, now we are also providing our in-vivo results in 3D volume rendered video format in MNI space to conveniently compare our unthresholded group average statistical maps. We present this video as a figure supplement to Figure 9 of our updated manuscript (Figure 9—video 1).

Thus, while we agree that CN and SOC present a lower level of SNR which affect in part the resulting overlap (and reliability) when compared to IC and MGB, we believe that a major reason for the lower overlap in these regions can be assigned to misalignment: there is not enough anatomical contrast to align the CN/SOC across participants. Furthermore, while we display the group probabilistic maps a threshold of 3 out of ten volunteers, both the CN and SOC our probabilistic maps have voxels where the overall across participants is up to 5 out of ten volunteers.

In conclusion, these results attest to the quality of the data and the validity of the analysis. We believe that our new analyses and figures show the reproducibility of our results, inspiring confidence in the reliability of the in vivo definitions also at the level of the CN and SOC.

I also wondered where the cluster threshold of 27 voxels came from (subsection “Functional MRI analysis”, last paragraph)?

In the text of the Materials and methods, we describe it as:

“applying of a cluster size threshold of 3.37 mm^3^ (27 voxels in the 0.5 mm isotropic anatomical space ~2.5 voxels in the original functional resolution).”

Later in the same paragraph, we state:

“The additional clustering allowed us to further control for possible false positives by imposing a neuroanatomically plausible hypothesis (i.e. none of our region of interest is smaller than 3.37 mm^3^ in volume).”

In more detail, we have decided on number `27` instead of `26` or `28` or others by thinking about the discrete cubical lattice that the data resides in. We have considered the following: a cube of 1 voxel, 2×2×2=8 voxels, 3×3×3=27 voxels, and 4×4×4=64 voxels. We deemed 64 voxels (2mm×2mm×2mm=8mm^3^) too big because one of our regions of interest (superior olivary complex) was reported to be around 6-7mm^3^. We have deemed 8 voxels (1mm^3^) and 1 voxel (0.125mm^3^) too small. Therefore we have decided to use 27 voxels.

And what the rest of the slices look like through the brainstem (e.g., how many splotches are there).

We agree with the reviewer that 2-D slices are inadequate for viewing 3-D clusters. In addition to Figure 4, we have therefore added 3-D animated functional representations, both at the group level and for individual participants (both thresholded and unthresholded). These animations are available as supplementary videos to the manuscript and at the links above.

2) The similarity between the estimates of CN and SOC across the BigBrain, ex-vivo, and in-vivo segmentations is near or at zero (as shown by the dice coefficients). Since defining these smaller structures is really the main potential contribution of the paper (given IC and MGB have been previously identified by a number of groups), this is concerning.

Image registration remains an open challenge, particularly across different image modalities (such as from histology to MRI, or between different MRI acquisition sequences). This issue is magnified in the brainstem, which has not generally been the focus of registration methods. Indeed, our BigBrain–MNI registration improvements highlight how important it is to pay attention to subcortical anatomy when assessing registration success. In Author response image 1 and Author response image 2 we show how, even with our registration improvement for overall specimen alignment, the borders and centroids of specific structures such as the inferior colliculi still differ between the BigBrain and post mortem MRI atlases, even when both have been nonlinearly registered to the MNI152 template.

**Author response image 1. respfig1:** Underlay image: BigBrain histology in MNI152 space. Yellow overlay: mask of post mortem anatomical MRI in MNI152 space.

**Author response image 2. respfig2:** Underlay image: BigBrain histology in MNI152 space. Segmentation overlay: conjunction of two raters’ IC segmentations based on post mortem anatomical MRI (conducted in native space) in MNI152 space.

Thus, when assessing the similarity between BigBrain and ex vivo MRI (which each have sufficient contrast to manually define the borders of the CN and SOC), the low similarity between the segmentations (compared to, for instance, the two independent segmentations of the BigBrain dataset) indicates that alignment is a critical factor to these metrics.

The in vivo data are additionally affected by: a) a further decrease in signal (and contrast) compared to the BigBrain and post mortem MRI samples; and b) partial voluming effects due to the lower spatial resolution.

These issues, which are discussed in the manuscript, affect the alignment of the in vivo data (especially in the CN/SOC regions). It has to be noted here that these data undergo two steps of alignment in our procedure: 1) to align all participants to one reference participant; 2) to align the average of all participants to the post mortem data.

The in vivo definition of the CN and SOC is reproducible both within and across participants as demonstrated by the new analyses reported in the manuscript (as well as above in response to an earlier comment). We believe these analyses address the reviewer concerns. Future work will have to address the possibility of improving the alignment of the CN/SOC both in vivo and between in vivo and post mortem data. As we discuss in the manuscript this requires the exploration of acquisition parameters that would allow the accurate anatomical definition of these structures in vivo.

3) The tractography results also essentially show that the known connections between any of the nuclei cannot be reliably reproduced, especially in-vivo. For instance, in the demonstration subject in Figure 6 as well as the matrix showing total streamlines, the IC is essentially being bypassed, despite the fact that it is (to my knowledge) the major source of input to the MGB.

While we agree that the results demonstrate how tractography methods are not yet perfect, we do show that the tractography results are consistent across participants (Figure 7—figure supplement 1). With regards to the IC, the streamline connectivity matrix does in fact show streamlines between the IC and the SOC and MGB on the same side (which are anatomically expected connections). Indeed, we can visualize streamlines hitting the IC from both below and above in the example participant tractography video, Author response video 2. We agree that the figure does not portray these streamlines clearly, so we added a 3-D rotation video to the online resources.

**Author response video 2. respvideo2:** In vivo probabilistic tractography of the subcortical auditory system in one individual.

However, it is true that the streamlines largely do not follow the neuroanatomically plausible path towards the IC, in contrast to the post mortem tractography. It is likely that the signal-to-noise ratio of in vivo DWI does not yet disambiguate the arcing white matter connections into the IC from both below and above. These lemniscal and brachial connections do appear in the post mortem tractography dataset, particularly as shown in Figure 4 and Figure 4—figure supplement 2 – the latter of which is downsampled to in vivo resolution, demonstrating that voxel sizes of approximately 1 mm should be adequate for identifying these connections with current analysis methods if the data has sufficiently high SNR.

In the Figure 3—figure supplement 1 of the postmortem data, there are also a lot of streamlines that must be incorrect (most prominently, the large set of streamlines going inferiorly, but also the huge projections from around the SOC and CN seeds that seem to bypass the IC and go directly into the MGB). The large projections going inferiorly from the IOC seeds (Figure 4) are also rather mysterious. Particularly given the statements regarding the 'vastly improved estimation of white matter connections'.

We appreciate the reviewer’s concerns regarding these figures. Indeed, we had debated how to represent the streamlines, originally deciding to show all streamlines that pass through auditory structures regardless of anatomical plausibility. However, since we do have a priori knowledge about the auditory system (such as its lack of connections with the spinal cord), it is reasonable to exclude streamlines that take anatomically implausible routes. Therefore, we have reanalyzed the tractography data to exclude streamlines that pass through the medulla, which includes the inferior-running streamlines mentioned by the reviewer.

In terms of overall contribution, having labelled nuclei in two publicly available postmortem datasets is helpful, and the fix to the incorrect MNI warp of bigbrain is also good to have. I'm less convinced about the contribution of the fMRI data (compared to the previously published studies that they come from), due to the way that they were processed and analyzed. If anything, the DTI tractography results seem to confirm the problems that Thomas et al., 2014, showed with even better diffusion data.

The in vivo fMRI data reported in the manuscript have never been published before and we have clarified some misunderstandings in the analysis pipeline. Our new analyses and data show the reproducibility of the in vivo definitions of lower auditory nuclei both within and between subjects. Our opinion is that these results are valuable in order to better understand past and future in vivo studies that have or will evaluate the functional properties of these regions. These analyses have originated from the helpful comments of this reviewer regarding our manuscript of which we are thankful.

Reviewer #3:This manuscript examines the structure of human auditory subcortical structures by recording in-vivo 7T functional magnetic resonance imaging (fMRI), structural MRI, and diffusion MRI (N=10). Ex-vivo 7T structural and diffusion MRI were also recorded in a single post-mortem subject. A subcortical atlas was created from the post-mortem structural MRI and publicly available histology data, and used to validate the functionally identified subcortical regions.All four subcortical auditory structures (MGN, IC, SOC, CN) were successfully identified using ex-vivo structural MRI, while in-vivo structural MRI could only be used to identify the MGN and IC. Using functional data, the MGN and IC could be identified in majority of the participants, while the SOC and CN could only be identified in a small subset participants (n = 2-3). Ex-vivo tractography showed significant within-structure streamlines while the number of between-structure streamlines was quite modest. On the other hand, in-vivo diffusion imaging revealed significant streamlines between and within structures.

We thank the reviewer for the comments, but we feel a clarification is needed on the summary above. The reviewer states that the SOC and CN could only be identified in 2 or 3 participants in the in vivo functional data. This is a misunderstanding caused by our choice to only present the functional probabilistic maps (across participants) in the original version of the manuscript. These maps color code voxels based on how many participant show a significant activation in response to sounds at that voxel. The lower probability in the CN and SCO (compared to IC and MGB) does not mean that the CN and SOC are significant only in 2 or 3 participants but rather could be due to the alignment across participants being of lower quality in these regions (as we discuss in the manuscript). In the original version of the manuscript we had described that all our participants showed significant responses at the level of CN and SOC. Now we substantiate this claim with new data, and several new analyses and figures. In particular, the new Figure 4 shows statistical maps in each individual as obtained in the experimental data already reported in the original manuscript.

This figure shows that the CN and SOC could be identified on the basis of fMRI data in all participants with the exception the right SOC in S03, the left CN and SOC in S08 and the left CN in S09. We also evaluated the reproducibility of these results by comparing functional definitions in the same participants across experiments (Figure 4—figure supplements 3 and 4). Also see 3D volume rendered videos that show both thresholded and unthresholded maps for all individual participants as reference in our new Figures 9, 10 and 11.

Thus, while future work will have to address the possibility of better aligning CN/SOC across participants (perhaps by exploring anatomical acquisitions in order to highlight these regions in vivo), we believe our data indicate that with sufficient statistical power CN and SOC can be identified in individual participants data.

In general, the paper makes a useful contribution to the literature, and the figures are lovely. The Introduction, however, could be reorganized to more clearly indicate the gap in knowledge and the research question that this work was designed to address, since the general location of the main subcortical auditory processing nuclei is already well known in humans.

We have reworked the Abstract, Introduction, and Discussion to make our contributions more clear. Foremost, we conclude the Introduction with the following take-home message:

“Where histology provides ground truth information about neural anatomy, we show that post mortem MRI can provide similarly useful three-dimensional anatomical information with less risk of tissue damage and warping. We also show that in vivo functional MRI can reliably identify the subcortical auditory structures within individuals, even across experiments. Overall, we found that each methodology successfully localized each of the small structures of the subcortical auditory system, and while known issues in image registration hindered direct comparisons between methodologies, each method provides complementary information about the human auditory pathway.”

Additionally, we added text about post mortem MRI in order to lay the groundwork for our novel use of this modality for investigating the human subcortical auditory system:

“Advances in MRI of the post mortem human brain allow for investigating three-dimensional (3-D) anatomy with increasingly high resolution (100 µm and below). This points to "magnetic resonance histology" (Johnson et al., 1993) as a promising avenue for identifying the small, deep subcortical auditory structures. However, to the best of our knowledge, post mortem MRI has not been utilized within the subcortical auditory system, although it has provided useful information about laminar structure in the auditory cortex (Wallace et al., 2016).”

The Abstract now concludes with:

“This work demonstrates the potential of functional MRI for investigating the human subcortical auditory system and contributes novel, openly available tools for researching the human auditory system to understand its structural organization.”

To open the Discussion, we have added:

“We showed that functional localization of the subcortical auditory system is achievable within each participant, and that localization is consistent across experimental sessions.”

One strength of the paper is that these findings replicate the authors' previous work (cited Discussion, ninth paragraph)-lending further credence to the idea that subcortical structures can be reliably identified from functional data. However, one limitation is that functional localization was not very strong in lower brainstem nuclei (i.e., SOC and CN) for majority of the participants (cf. Figure 5 left and middle panel and Figure 5—figure supplement 1).

We agree that the lower brainstem nuclei are more challenging to identify due to their small size and deep location. However, as we have already clarified above, we were able to identify all main auditory nuclei in each of our subjects; we have added Figure 5 demonstrating these findings in every participant. In a supplementary figure we also demonstrate the reproducibility of this localization within six of the original ten participants that took part to a new experiment.

It is only when combined in a common anatomical space that the results look less strong. We believe that this arises in large part due to ongoing challenges regarding image registration, particularly in lower brainstem regions that are discussed in the revised manuscript.

I do wonder if the in-vivo portion of the study is under powered. This issue could be addressed using additional commentary by the authors, although if the N is not supported by a power analysis (or some other logical explanation), the paper could be improved by recording data from a larger sample.

We thank the reviewer for this comment that highlights the need for more clarity about our study design. As we now clearly state in the manuscript, this study was tailored to obtain a functional definition of the auditory pathway in each individual (and not taking advantage of group analyses), as we already had evidence from previous work in the IC/MGB that anatomical variability would have been a major issue in a potential group study. We now state that:

“Leveraging the increased SNR available at 7Ts, we aimed to collect data that would allow a functional definition of the auditory subcortical auditory structures in single individual participants. For this reason, we collected a large quantity of data in all individuals (2 sessions with 12 runs each in Experiment 1 and 2 sessions with 8 runs each in Experiment 2), and all statistical analyses were performed at the single subject level. Group analyses were used to evaluate the correspondence across subjects of individually defined regions (i.e., the definition of a probabilistic atlas across participants) as well as the ability of generalizing to new participants by means of a leave-one-subject-out analysis.”

The amount of data necessary to identify these nuclei functionally in an individual was piloted by acquiring data in one session for one individual, as prior data do not exist that would allow for the definition of the statistical power attainable at 7T in regions such as the CN and SOC. Preliminary analysis of the single subject suggested that a single session’s worth of fMRI data was too noisy to specifically identify the auditory structures. We then added a second functional MRI session, doubling the data and providing ample SNR for auditory localization.

We were able to functionally localize the subcortical auditory structures in every participant. To further augment the reliability of our findings, we now include a second study including 6 of 10 subjects who participated in Experiment 1. Figure 4—figure supplement 3 shows similar spatial activation patterns between experiments for most participants.

We believe that these results demonstrate the efficacy of our acquisition methods for localizing brainstem auditory structures within individuals that are reliable across scanning sessions.

The group analysis that follows the definition of the auditory nuclei in each participants allowed us to highlight the difference in anatomical registration that can be obtained across structures in the auditory pathway. Compared to our previous work (Moerel et al., 2015), we were now able to perform one anatomical alignment procedure for the whole brainstem. This procedure produced satisfactory results for the IC/MGB (improving on the separate alignment procedure followed in (Moerel et al., 2015) but requires further development for the CN/SOC. We believe this is mostly due to the lack of anatomical contrast highlighting these regions in each individual and highlight this as potential future work in the revised manuscript.